# The role of aerosols and meteorological conditions in shaping cloud droplet development in New Mexico summer deep-convective systems

Huihui Wu[1*], Nicholas Marsden[2], Paul Connolly[1], Michael Flynn[1], Paul I. Williams[2], Declan Finney[3], Kezhen Hu[1], Graeme J. Nott[4], Navaneeth Thamban[1], Keith Bower[1], Alan Blyth[3], Martin Gallagher[1] and Hugh Coe[1,2]

[1]Department of Earth and Environmental Sciences, University of Manchester, Manchester, UK
[2]National Centre for Atmospheric Science, University of Manchester, Manchester, UK
[3]School of Earth & Environment, University of Leeds, Leeds, UK
[4]FAAM Airborne Laboratory, Cranfield, UK
*Now at: Univ Paris Est Créteil and Université Paris Cité, CNRS, LISA, 94010 Créteil, France

*Correspondence to*: Hugh Coe (hugh.coe@manchester.ac.uk)

**Abstract.** The accurate representation of aerosol physicochemical properties is important when describing aerosol-cloud interactions. The Deep Convective Microphysics Experiment (DCMEX) aimed to improve the representation of microphysical processes in deep convective systems. As part of this project, an airborne campaign (July to August 2022) was conducted to characterize the thermodynamics-dynamic-aerosol-cloud system over the isolated Magdalena Mountains in New Mexico, US. Backward dispersion analyses identified a transition in dominant airmass origins from Northwest (NW) continental to Southeast (SE) aged oceanic flow, coinciding with substantial changes in meteorological conditions and aerosol characteristics. The SE-flow period generally exhibited lower lifting condensation levels, enhanced convection, higher boundary layer humidity, and more frequent and intense precipitation, compared to the NW-flow period. During the SE-flow period, aerosol size distributions showed a more pronounced bimodality, characterized by increased Aitken-mode concentrations and fewer but larger accumulation-mode particles, indicating enhanced cloud processing. Submicron aerosols exhibited larger sulfate fractions, more oxidized organics, and greater hygroscopicity. Correspondingly, clouds presented larger droplet sizes and liquid water contents. A bin-microphysics parcel model was also employed to simulate the development of cloud droplets, constrained by airborne observations. Model-observation comparisons highlight the critical role of aerosol entrainment in reproducing the observed broad cloud droplet spectra extending toward small sizes. The results underscore the combined influence of meteorological conditions and aerosol characteristics on cloud microphysics under different flow regimes and emphasize the importance of aerosol entrainment in the development of deep convective clouds. This study provides valuable constraints for improving parameterizations of aerosol-cloud interactions in convective systems.

## 1 Introduction

Deep convective clouds play a crucial role in driving precipitation patterns, the hydrological cycle, and large-scale circulation (Houze, 2014). These clouds also influence the regional and global radiation budget by reflecting incoming solar

radiation and absorbing terrestrial longwave radiation (Jensen and Del Genio, 2003). Furthermore, deep convective cloud systems are likely associated with extreme weather events such as severe thunderstorms, tornadoes, and flash floods, which result in global economic losses (Newman and Noy, 2023). Given their significant effects on the climate system, the accurate representation of deep convective clouds and their microphysical processes in models remains a long-standing challenge for improving weather and climate predictions.

The life cycle of deep convective clouds is modulated by complex microphysical processes, including droplet formation, droplet growth through condensation and coalescence, thermodynamic phase transitions between liquid droplets and ice crystals, and the development of precipitation (Arakawa, 2004). Aerosols play a key role in these processes, and aerosol-cloud interactions are considered among the largest uncertainties in estimating climate sensitivity to radiative forcing (Boucher et al., 2013; Forster et al., 2021). Aerosols can affect clouds by acting as cloud condensation nuclei (CCN) or ice nucleating particles (INP), which are termed aerosol indirect effects (Boucher et al., 2013). In the presence of aerosols that are sufficiently effective as CCN, water vapor condenses onto CCN surfaces to form cloud droplets. This initial process in the lifecycle of convective clouds mostly occurs at cloud base (Tao et al., 2012; Rosenfeld et al., 2014). Generally, higher CCN concentrations produce a greater number of smaller droplets and narrower droplet size distributions, which are likely to inhibit collision-coalescence and delay raindrop formation, thereby extending cloud lifetime (Rosenfeld, 2000; Tao et al., 2012). This delay can have opposing effects on convective cloud development: the increased condensational heat release tends to enhance cloud buoyancy and vertical development, while the resulting increase in condensate loadings partially offsets that buoyancy enhancement (Rosenfeld et al., 2008; Koren et al., 2014; Fan et al., 2018; Varble et al., 2023). However, the presence of giant CCN, such as coarse-mode sea salt aerosols, can produce initially large droplets and accelerate warm rain formation, thereby inhibiting the vertical development of convective clouds (Yin et al., 2024). These existing studies suggest that the CCN ability of aerosols determines the initial droplet number concentration and size distributions, thereby influencing subsequent cloud dynamics throughout convective cloud lifetime. Additionally, INPs can promote the heterogeneous freezing, and regulate ice crystal number concentrations during convective development (Tao et al., 2012). Detailed characterization of aerosol amounts and properties is therefore crucial for improving the representation of aerosol-cloud interactions in atmospheric models, in particular, aerosol size distribution and chemical composition which determine their CCN and INP ability (Petters and Kreidenweis, 2007). However, the representation of aerosol properties and associated indirect effects is complex and uncertain, as they are subject to atmospheric dynamic and thermodynamic conditions (Yang et al., 2020). Atmospheric transport generates temporally varying aerosol types, amounts, and properties in a given region (Yang et al., 2020). Moreover, cloud response to aerosol perturbations depends on environmental conditions, such as cloud-base temperature, updraft velocity, and humidity (Pruppacher and Klett, 2010). Previous studies have indicated that clouds with cool, high bases tend to exhibit little sensitivity in cloud-top height and precipitation to aerosol loadings, while clouds with warm, low bases display larger aerosol-induced changes (e.g. Li et al., 2011). Overall, it is critical to understand how varying aerosol properties influence convective cloud microphysics under different environmental conditions.

Another key uncertainty in understanding the development of convective clouds is the role of entrainment and mixing. Theoretically, cloud-droplet growth in a closed (adiabatic) parcel leads to narrower size distributions as vertical development progresses, tending to suppress the onset of coalescence through differential gravitational sedimentation (Pruppacher and Klett, 2010). However, observations have revealed that cloud-droplet size distributions are relatively broader than those in ideal adiabatic parcels, and this result is usually attributed to a consequence of entrainment and mixing and secondary activation

(Blyth, 1993; Chandrakar et al., 2016). Despite its recognized importance, the representation of the entrainment and mixing process in models remains uncertain. It is suggested that inhomogeneous mixing typically dominates when cloud droplets are small, as their evaporation rates significantly exceed the mixing rate of clouds with surrounding subsaturated air (Pruppacher and Klett, 2010). In this way, a subset of cloud droplets evaporates completely, leaving the others in the volume unchanged. When cloud droplets are larger and their evaporation rates are comparable to the mixing rate, homogeneous mixing dominates

the system and the influence of inhomogeneous mixing weakens (Pruppacher and Klett, 2010). With homogeneous mixing, droplets evaporate by uniformly reducing their size across the population, leaving droplet number density largely unchanged except through simple dilution. While some studies suggest that entrainment and mixing in convective clouds are almost completely inhomogeneous (e.g. Burnet and Brenguier, 2007; Braga et al., 2017a), some other studies propose that inhomogeneous mixing may dominate early cloud development when droplets are small, and then homogeneous mixing may

become more prevalent as convective clouds evolve (e.g. Lehmann et al., 2009). It is unclear whether the entrainment is predominantly homogeneous mixing, inhomogeneous mixing, or a combination of both. An improved understanding of how these mixing mechanisms dominate throughout the development of convective clouds is essential for improving the representation of their microphysical processes. Moreover, the consequence of entrainment processes on cloud microphysical evolution can be regulated by surrounding environmental conditions, such as relative humidity (RH) and aerosol characteristics

(Koren et al., 2010). In particular, aerosols entrained with surrounding air can act as additional CCN and promote secondary activation of droplets above cloud base. Such additional activation has been identified as an important factor contributing to the broadening of droplet size distributions, toward small droplet diameters (Lehmann et al., 2009; Chandrakar et al., 2016). An improved understanding of aerosol entrainment will reduce the current uncertainty in predicting droplet number concentrations and size distributions.

Overall, existing studies show that the impacts of aerosols, meteorological conditions, and entrainment processes on the formation and development of deep convective clouds are highly complex, which introduces large uncertainties in understanding the climate effects associated with deep convective systems. Advancing understanding in this area requires coordinated observational and modeling efforts across diverse convective regimes with different aerosol and meteorological conditions. To improve the representation of aerosol-cloud interactions and cloud microphysical processes in deep convective

systems, the Deep Convective Microphysics Experiment (DCMEX, July to August 2022) was conducted over the Magdalena Mountains, New Mexico (local time = UTC – 6 hours). Isolated convective clouds can frequently form and develop over these mountains during the summer season in North America, providing ideal laboratory-like conditions for investigating deep convective processes. Earlier studies primarily characterized convective clouds and precipitation development in this area (i.e.

Blyth and Latham, 1993). The DCMEX campaign provided coordinated and comprehensive observations of aerosols, cloud microphysics, and thermodynamics of the deep convective system over the Magdalena Mountains (Finney et al., 2024). The campaign benefits from significant advances in measurement techniques, including high-resolution cloud probes with improved particle sizing and phase discrimination, and state-of-the-art aerosol instruments capable of resolving physicochemical properties (Finney et al., 2024). In this study, we focus on the influence of aerosol properties, meteorological conditions and entrainment processes on the formation and development of cloud droplets during the campaign, combining observations with bin-microphysics parcel model simulations. This study aims to improve the representation of microphysical processes in this region and similar deep convective systems in future model studies.

## 2 Methods

During the DCMEX field campaign, airborne measurements were carried out using the UK FAAM (Facility for Airborne Atmospheric Measurements) Airborne Laboratory (BAe-146 Atmospheric Research Aircraft), with flight operations centred over the Magdalena Mountains (~33.98°N, ~107.18°W), New Mexico, US. In addition to airborne measurements, Doppler radars, automated cameras, and ground-based aerosol and meteorology measurements were also deployed during the field campaign. More details on the DCMEX project are provided by Finney et al. (2024). This study focuses on airborne measurements, supplemented by observations from meteorological stations installed at Langmuir Laboratory on the summit of the Magdalena Mountain Range (elevation: ~3240 m above sea level (a.s.l)). A total of 19 scientific flights (designated flight labels from C297 to C315) were performed between July 16 and August 8, 2022. Figure 1a shows flight tracks during the campaign. Each flight took off from Albuquerque International Sunport and normally started with a profile ascent to 8–9 km a.s.l, where a dropsonde (Vaisala RD41) was deployed in the vicinity of the Magdalena Mountains. The flights included a series of straight and level runs (SLRs) for aerosol characterization, which was performed around the mountains prior to the initiation of convective cloud cells and followed a designated kite-shaped pattern (solid red kite in Fig. 1a). The aerosol SLRs were conducted at various altitudes, including low-level, close to cloud-base and high-level runs. Cloud passes typically followed the aerosol runs, allowing for the sampling of developing convective clouds over the Magdalena Mountains at altitudes ranging from close to cloud base to approximately the −20°C isotherm. In some cases, cloud passes were also performed slightly away from the mountains. The sampling strategy focused on early to mid-development stages of convective clouds with supercooled tops. Mature or dissipating clouds were generally avoided or departed from, because of their lower likelihood of containing significant supercooled water, and stronger turbulence hazards for safe aircraft operations. As a result, complete glaciation was rarely encountered within the penetrated clouds, even though it occurred in maturing convective elements in the region. The FAAM Bae-146 was equipped with a suite of instruments to measure atmospheric dynamics and thermodynamics, including such as wind speed and direction, temperature (T), and RH. A brief description of onboard aerosol (number concentrations, size distributions, and submicron chemical compositions) and cloud (number concentrations and size

distributions) measurements used in this study are provided in Sect. 2.1. Ground-based meteorological stations provided observations of surface T, RH, pressure, and precipitation intensity.

## 2.1 Instrumentation

Submicron aerosol chemical compositions were measured using a compact time-of-flight aerosol mass spectrometer (C-ToF-AMS, Aerodyne Research Inc, Billerica, MA, USA) and a Single Particle Soot Photometer (SP2, Droplet Measurement Technologies, Boulder, CO, USA). The AMS can provide online chemical characterization across a range of ion mass-to-charge (m/z) ratios from 10 to 500 (Drewnick et al., 2005). The AMS was calibrated using monodisperse ammonium nitrate and ammonium sulfate particles after each flight. The AMS raw data were processed using the standard SQUIRREL (SeQUential Igor data RetRiEvaL, v.1.60N) TOF-AMS software package. A time- and composition-dependent collection efficiency was applied to the data based on the algorithm by Middlebrook et al. (2012). In this study, the mass concentrations of organics, sulfate, nitrate, and ammonium were determined. Organic fragment marker (m/z 44) was used to estimate the oxygen-to-carbon (O:C) ratio of organics following the methods developed by Aiken et al. (2008). The SP2 can measure the refractory black carbon (hereafter referred to as BC) containing particles with an equivalent spherical diameter in the range of 70–850 nm. It can provide BC mass, size, and mixing state. The SP2 incandescence signal, which is proportional to the mass of BC present in the particle, was calibrated using Aquadag BC standards (Aqueous Deflocculated Acheson Graphite, manufactured by Acheson Inc., USA). Details on the AMS and SP2, including instrument principles as well as operation, calibration, and data interpretation for aircraft deployment have been described previously (Drewnick et al., 2005; Liu et al., 2010; Morgan et al., 2010).

Aerosol size distributions were measured via an on-board scanning mobility particle sizer (SMPS) and a wing-mounted passive cavity aerosol spectrometer probe (PCASP). The SMPS connects a low-pressure water-based condensation particle counter (WCPC model 3786-LP) to a TSI 3081 differential mobility analyser (DMA). It provides aerosol size distribution within the mobility diameter range of 20 to 350 nm, across 40 logarithmically spaced bins. The SMPS data were inverted using the scheme developed by Zhou (2001), based on a ~1 min averaging time. Given the low time resolution, SMPS data were only available during SLRs with relatively stable aerosol levels. The PCASP resolved number concentrations at 5-Hz resolution, across 30 diameter bins between 0.1 and 3 μm. The PCASP was calibrated using di-ethyl-hexylsebacate (DEHS) and polystyrene latex spheres (PSL) with known size and refractive index (Rosenberg et al., 2012). Mie scattering theory was used to determine the bin sizes by assuming particles are spherical, with a refractive index of $1.59 - 0.024i$. Aerosol number concentrations in the accumulation mode ($N_a$, $0.1 - 3$ μm) were obtained by integrating the PCASP size distributions. A model 3786-LP water-filled condensation particle counter (CPC) on board can detect aerosol particles larger than 2.5 nm (Hering et al., 2005) at 10-Hz resolution, and it can provide number concentrations of all submicron aerosol particles including ultrafine particles ($N_{sa}$, 2.5 nm to 1 μm) at an accuracy of $\pm 12$ %.

Cloud droplet number concentrations and size distributions were measured using a Cloud Droplet Probe (CDP), following the operation and calibration procedures in Lance et al. (2010). The CDP measures the forward scattered light over a 1.7 to 14°

solid angle when cloud droplets pass through its incident laser beam. Cloud droplet particles are detected at 25-Hz resolution, and in 30 bins over a nominal size range of 2–50 μm. The size calibrations were performed during pre-flight, using 10 different-
size glass beads with known diameters and refractive index (Rosenberg et al., 2012; Barrett et al., 2022). The cloud droplet number concentration ($N_d$), cloud droplet effective radius ($R_e$), and liquid water content (LWC) were calculated from the CDP's cloud droplet spectrum as follows:

$$N_d = \int n(r) \, dr \approx \sum_1^m n(r_i) \tag{1}$$

$$R_e = \frac{\int r^3 \, n(r) \, dr}{\int r^2 \, n(r) \, dr} \approx \frac{\sum_1^m r_i^3 \, n(r_i)}{\sum_1^m r_i^2 \, n(r_i)} \tag{2}$$

$$LWC = \frac{4\pi}{3} \rho_{water} \int r^3 \, n(r) \, dr \approx \frac{4\pi}{3} \rho_{water} \sum_1^m r_i^3 \, n(r_i) \tag{3}$$

where $n(r_i)$ is the number of cloud droplets in a particular size bin, $r_i$ is the middle radius value for each of the size bins, and $\rho_{water}$ is the density of liquid water. An LWC value of 0.01 g m$^{-3}$ was used to define the low threshold for the presence of cloud. To eliminate the inclusion of optically thinner clouds, thresholds of $N_d > 5$ cm$^{-3}$ and bulk LWC from CDP $> 0.02$ g m$^{-3}$ were used to perform statistical cloud sample analysis.

Table S1 shows a summary of data availability for aerosol and cloud measurements in this study. All aerosol (AMS, SP2, SMPS, PCASP, CPC) and cloud (CDP) measurements were corrected to standard temperature and pressure (STP) (273.15 K and 1013.25 hPa).

**2.2 Backward dispersion simulations**

        To investigate the origin and transport pathways of airmasses over the Magdalena Mountains, backward dispersion
simulations were performed using the UK Met Office's Numerical Atmospheric Modeling Environment (NAME) (Jones et al., 2007). NAME was selected for this study due to its ability to utilize high-resolution meteorological data, with a grid resolution of approximately 10 km × 10 km. It is capable of predicting dispersion over a wide range of spatial scales, from a few kilometers to global distances. For each simulation, a specified number of hypothetical tracer particles were released from a 0.2° × 0.2° grid box centered over the Magdalena Mountains (34.0° N, 107.2° W). Low-level and high-level simulations were
performed for each flight case separately. Low-level simulations released tracer particles during the time when low-level and below-cloud aerosol samplings were conducted, while high-level simulations released particles during high-altitude aerosol samplings. The trajectories of released tracer particles were traced backward for 6-days, based on three-dimensional gridded (3D) meteorological fields from the UK Met Office's Unified Model (MetUM) (Brown et al., 2012). These meteorological fields were updated every three hours, with a high horizontal resolution of 0.14° longitude by 0.1° latitude and 59 vertical
levels extending up to ~ 29 km in altitude. In this study, the integrated horizontal footprints were output for each simulation, representing the cumulative air parcels passing through each grid over the past 6-days. The footprints are expressed in units of mass concentration (g m$^{-3}$), based on the predetermined quantity of released air parcels.

## 2.3 Bin-microphysics parcel model

Simulations of cloud parcel development were conducted using a bin microphysics parcel model (https://github.com/UoM-maul1609/bin-microphysics-model, University of Manchester), an adapted version of the model in Fowler et al. (2020). The bin microphysics model simulates aerosol activation into cloud droplets followed by processes such as condensation and/or deposition from water vapour, collision and coalescence of cloud droplets, and both primary and secondary ice formation processes (James et al., 2023). This study focuses on processes related to the formation and growth of cloud droplets. In the model, aerosol particles are represented by multiple lognormal modes, with each mode treated as an internal mixture with the same chemical composition. The lognormal size distributions of aerosols for each mode are as follows:

$$\frac{dN}{d\ln D_a} = \frac{N_T}{\sqrt{2\pi}\ln\sigma_g}\exp\left[-\frac{\ln^2\left(\frac{D_a}{D_{a,m}}\right)}{2\ln^2\sigma_g}\right] \tag{4}$$

where N is the number concentration of aerosol particles, $D_a$ is the aerosol particle diameter in the mode, $N_T$ is the total number of aerosol particles in the mode, $D_{a,m}$ is the median aerosol particle diameter for the mode, and $\sigma_g$ is the geometric standard deviation of the logarithmic distribution. The model is initialized by summing multiple lognormal size distributions.

The activation of aerosol particles is determined by the condensation of water vapor onto aerosol particles, following the equilibrium vapor pressure described by the κ–Köhler theory, which relates aerosol size and hygroscopicity through a single parameter κ (Petters and Kreidenweis, 2007). The subsequent growth of cloud droplets by diffusion accounts for mass accommodation, which is adjusted by modified diffusivity and conductivity (Jacobson, 2005; Pruppacher and Klett, 2010). Initial cloud droplet growth is governed by the diffusional growth equation, and further growth into raindrops is driven by the collision-coalescence process. The collision-coalescence process is modeled based on the stochastic collection equation using Bott's (1998) method, with collisional efficiencies derived from the Long (1974) kernel. The model employs a binned distribution to represent the size spectrum of liquid droplets, and particles that exceed a diameter of 3 μm are defined as droplets. In the simulations, the initial updraft velocity was specified, with adjustments to account for the effects of buoyancy on the updraft momentum.

In convective clouds, entrainment mixing is the process of incorporating surrounding environmental air into a rising plume. In this study, the simulation of the entrainment process, including both homogenous and inhomogeneous mixing, follows the theoretical framework outlined in Pruppacher and Klett (2010). In the model, the entrainment rate is defined by the ratio of the entrainment parameter (C=0.20) to the radius of the parcel. The air entering the parcel is calculated from thermodynamic profiles of potential temperature and water mixing ratio derived from dropsondes. For homogeneous mixing, entrained subsaturated air uniformly mixes with the cloud parcel, leading to a uniform reduction in droplet size. Inhomogeneous mixing follows similar equations but incorporates different treatments of droplet evaporation and number concentration adjustment (Baker, Corbin, and Latham, 1980). For inhomogeneous mixing, entrained subsaturated air remains in localized pockets rather than mixing instantaneously, leading to the evaporation of some droplets while others out of the pockets are unaffected. This process preserves the initial droplet size distribution but reduces the droplet number concentration.

The droplet number concentrations in each size bin are adjusted to conserve the parcel humidity, preventing a uniform reduction in droplet size as in the homogeneous mixing. Additionally, inhomogeneous mixing retains aerosols released from evaporated droplets in the discrete packets of subsaturated air, within the parcel, which may become re-activated later. More details in the simulations with the entrainment process are given in the supplementary and Sect. 3.3.

## 3 Results and discussion

### 3.1 Airmass source and meteorological conditions

NAME backward-dispersion fields are used to represent the horizontal footprint of original air parcels transported over the past six days before arriving at the sampling area over the Magdalena Mountains. Figure S1 provides some examples of backward-dispersion fields from selected flight simulations, illustrating different and representative transport pathways of source air parcels before reaching the sampling area during the campaign. In this study, the dispersion regions are classified
into four different regions, as shown in Fig. 1a. Figure 1b shows the fractional contributions of different source regions to the air parcels arriving at the sampling area throughout the field experiment, differentiating between low-level (bottom panel) and high-level (top panel) simulations. At the beginning of the campaign from July 16 to 23 (flights C297 to C301), NAME simulations suggest that the majority of air parcels originated from the northwest (NW) area of the Magdalena Mountains. Original air parcels from California were initially transported to the northeast (NE) area before shifting westward (Fig. S1a)
or directly moved eastward to the Magdalena Mountains (Fig. S1b), both of which were associated with continental flow. A small fraction of original air parcels arrived from the southeast (SE) area of the Magdalena Mountains, likely carrying aged oceanic airmass from the Gulf of Mexico. Additionally, low-level simulations generally indicate a slightly greater influence of SE-flow compared to high-level simulations. On July 24 (flight C302), NAME simulations indicate a distinct mixture of airmass sources originating from both the NW and SE of the Magdalena Mountains (Fig. S1c). From July 25 to August 1
(flights C303 to C309), the airmass source shifted predominantly to the SE region, with air parcels moving across the Gulf of Mexico and Texas regions (Figs. 1b, S1d, and S1e). Although the long-range airmass sources from August 2 to 4 (flights C310 to C312) remained primarily from the Gulf of Mexico, original air parcels were initially transported to the southwest (SW) and NW areas of the Magdalena Mountains (Arizona and California area) before moving south and eastward (Fig. S1f). This pattern was driven by a temporary shift in local winds, transitioning from southerly to northerly flow around August 3, which
returned to southerly before the end of the campaign (Finney et al., 2024). The subsequent days from August 6 to 8 (flights C313-C315) followed a similar source pattern to earlier days (July 25 to August 1, flights C303 to C309) which were dominated by the SE oceanic flow, but with more contributions from the NE continental flow (Fig. S1g).

For deep-convective cloud systems, the lifting condensation level (LCL) height is considered as a reasonable approximation of cloud base height (Finney et al., 2024). The lifting condensation level temperature ($T_{LCL}$) and LCL height
were calculated using the functions in MetPy Python Package (May et al., 2022), based on the observations of surface T, dewpoint temperature, and pressure from the Magdalena Observatory (Finney et al., 2024). The estimates were averaged over

the 15:00–16:00 UTC period, to represent the conditions prior to convection. Figure 2a shows the time series of the estimated $T_{LCL}$ and LCL heights during the DCMEX campaign. At the beginning of the campaign, LCL heights remained relatively high and $T_{LCL}$ remained relatively low. From July 23, LCL heights decreased and $T_{LCL}$ increased substantially. Figure 2b shows the estimated convective available potential energy (CAPE), based on the profiles of T and pressure from dropsonde observations in each flight. CAPE was calculated using the function in MetPy Python Package (May et al., 2022). During the early stage of the campaign (July 16 to 24, flights C297-C302), CAPE values were relatively low, ranging from about 20 to 190 J kg$^{-1}$. From July 25 (C303 to C309), CAPE increased to an approximate range of 280 to 680 J kg$^{-1}$, before a general decrease from August 2 (flight C310) onwards. CAPE is suggested to be an indicator of atmospheric instability and the potential for thunderstorm development (Stull., 2016). The observed general increase in CAPE from July 25 (flight C303) occurred with the shift in dominant airmass source from NW to SE, suggesting a higher probability of stronger convection during the SE-flow period. Observations also showed that the SE-flow period had more frequent lightning flashes and higher maximum cloud-top heights over the Magdalena Mountains (Finney et al., 2024), which is consistent with the stronger convection indicated by CAPE. Figures 2c and 2d show the time series of surface RH and T observed at Langmuir Laboratory, along with the corresponding RH and T within 200 m below the LCL height to represent close to cloud-base conditions for each flight. Surface observations generally showed lower RH and higher T during the NW-flow period (prior to July 24) compared to the SE-flow period. As LCL height decreased with the shift in dominant airmass source, the SE-flow period generally had higher T and RH near the LCL (close to cloud-base) compared to the NW-flow period. Figure 2e shows the time series of rain intensity (mm/hr) observed at the Langmuir Laboratory during the campaign. An increase in the frequency and intensity of precipitation was observed from July 23 (C301).

In summary, airmass sources and meteorological conditions in the sampling area exhibited variability throughout the campaign. Two distinct periods are identified, as illustrated in Fig. 1b. During Period 1 (July 19 to 23, flights C297–C301), at the beginning of the campaign, the air mass sources were primarily influenced by NW and/or NE continental flow. This period was generally characterized by relatively weak convection, low LCL heights, low surface RH, and high surface T. During Period 2 (July 25 to August 1 and August 6 to 8, flights C303–C309 and C313–C315), the dominant air mass source transitioned to SE oceanic flow. Compared to Period 1, Period 2 was generally associated with enhanced convection, higher LCL heights, higher surface RH, and lower surface T. This pattern reflects the warmer, drier airmass mainly transported by continental flow from the NW and the cooler, moister airmass originating from the Gulf of Mexico with SE flow. Additionally, there were intermediate days (flights C302 and C310–C312) characterized by a mixture of continental and oceanic air mass sources. The observed variability indicates a potential influence of the source airmass change on meteorological conditions in this region.

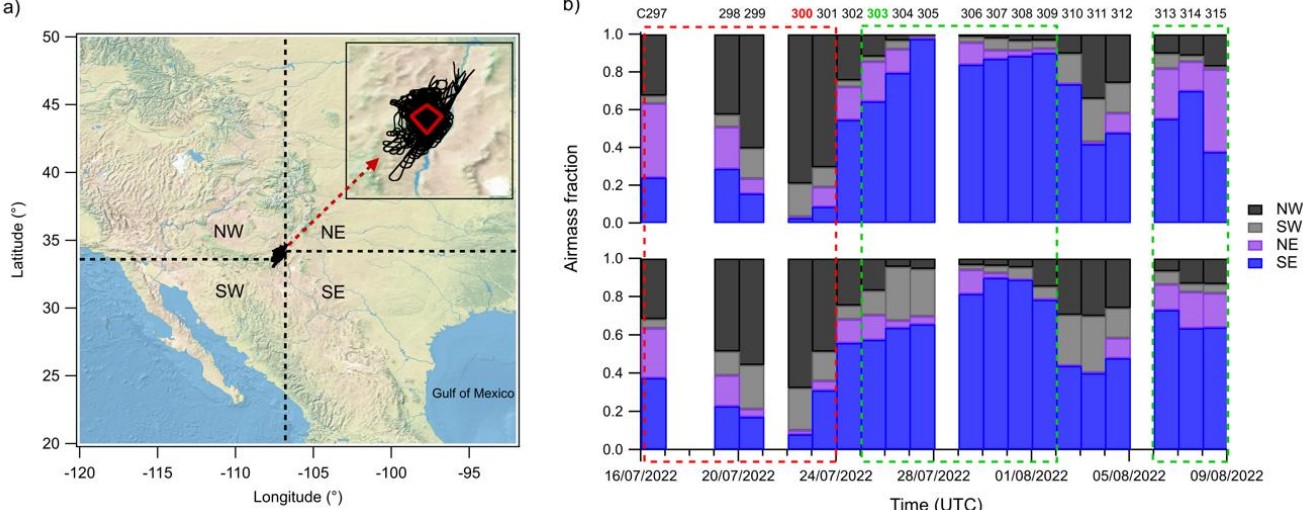

**Figure 1. a) Flight tracks during the DCMEX aircraft campaign (designated flight labels from C297 to C315). Each flight followed a designated kite-shaped pattern (red solid lines) over the Magdalena Mountains, New Mexico, USA. Black dashed lines represent four different regions used for air mass source classification. b) Fractional contributions of different source regions to the air parcels arriving at the sampling area throughout the field experiment. The bottom plot represents low-level simulations, and the upper plot represents high-level simulations. The dashed red box represents flights included in Period 1, and the green boxes represent flights included in Period 2. The corresponding flight numbers are indicated in the plot, and the two flights (C300 and C303) used for case simulations with bin microphysics parcel model are highlighted in red and green colours.**

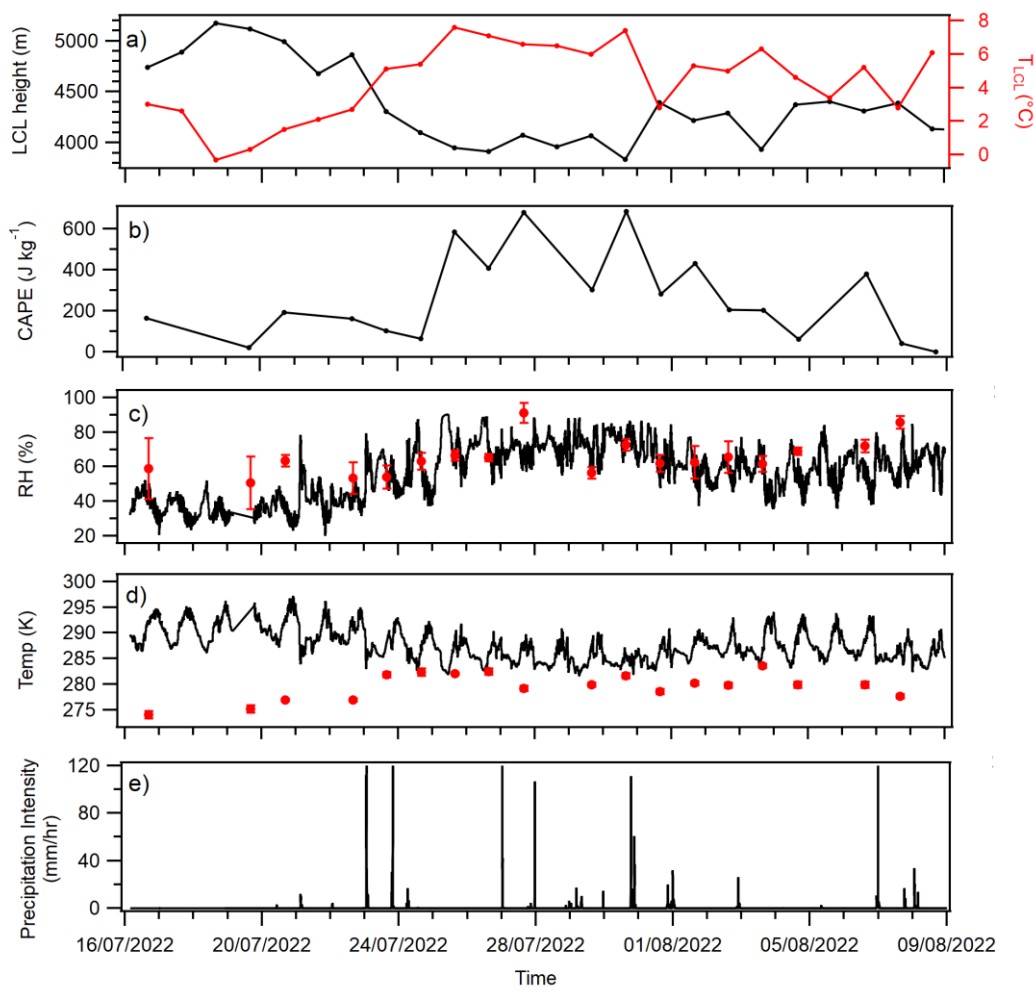

**Figure 2. From the top to the bottom: a) Daily LCL (left axis) height and $T_{LCL}$ (right axis) during the DCMEX campaign, which is estimated from Magdalena observations (15:00–16:00 UTC period). b) The calculated convective available potential energy (CAPE), based on dropsonde observations in each flight. c, d) Time series of surface RH and T observed at Langmuir Laboratory, along with the corresponding near cloud-base RH and T for each flight, the red markers and error bars represent averages and standard deviations within 200 m below the LCL height. e) Time series of precipitation intensity observed at Langmuir Laboratory.**

### 3.2 Aerosol Properties

### 3.2.1 Vertical profile of aerosol concentration

Figures 3a and 3b show the vertical profiles of out-of-cloud aerosol number concentrations from PCASP measurements ($N_a$) and out-of-cloud submicron aerosol number concentrations from CPC measurements ($N_{sa}$, 2.5 nm to 1 µm), respectively, for each flight. It should be noted that the y-axis represents the relative altitude with respect to the LCL height, which is an approximation of cloud base height in this study. The corresponding vertical profiles of aerosol number concentrations in terms of absolute altitude (above sea level, a.s.l.) are shown in Fig. S2.

At the beginning of the campaign (July 16 to 24, flights C297-C302), aerosols (0.1 to 3 μm) were observed from the surface up to altitudes above the LCL, with relatively constant and high $N_a$ reaching 6–6.5 km a.s.l., followed by a steep decline

at higher levels. In the middle of the campaign (July 25 to August 1, flights C303-C309), aerosols (0.1 to 3 μm) were mainly concentrated within the LCL layer. Above the LCL, $N_a$ monotonically decreased with altitude, with minimal amounts observed above 5.5–6 km a.s.l. For the later flights (from August 2, flight C310 onwards), relatively constant $N_a$ generally extended again from the surface up to altitudes slightly above the LCL. Aerosols larger than 0.1 μm are suggested to behave as the dominant CCN in most environments (Dusek et al., 2006; Braga et al., 2022). In this deep-convective cloud system, aerosols

larger than 0.1 μm (or CCN) can be activated to produce cloud droplets at/above the cloud base and subsequently removed from the atmosphere through following precipitation, a process known as in-cloud scavenging (rainout). The substantial decrease in $N_a$ above the LCL after July 24 is possibly due to the more efficient in-cloud scavenging on these days, which is indicated by the increase in the frequency and intensity of precipitation from July 23 (Fig. 2e).

The vertical structures of $N_{sa}$ during the campaign followed a temporal evolution pattern similar to that of $N_a$. However,

an enhancement in $N_{sa}$ was observed at high altitudes (> 7 km a.s.l.) in some flights, such as C303-C305. This enhancement of $N_{sa}$ but not $N_a$ suggests the presence of small particles at high altitudes. Similar findings have been reported in other convective regions, including the Pacific and Atlantic oceans and the Amazon Basin (Andreae et al., 2018; Williamson et al., 2019). It is suggested that deep convective clouds are able to transport condensable vapours (such as low-volatility organic compounds and sulfuric acid) into the upper troposphere, where new particle formation (NPF) from detrained

condensable vapours may contribute to a large number of ultrafine particles (particles smaller than 50 nm) (Williamson et al., 2019). In addition, deep convective clouds can remove pre-existing large particles that would act as sinks for small particles and condensable vapours, thereby further enabling the occurrence of NPF at high altitudes (Ekman et al., 2006). Recent studies over the Amazon have highlighted the important role of extremely low-volatility organic compounds (e.g. Isoprene-derived organonitrates) formed from the oxidation of biogenic volatile organic compounds (VOCs) in driving this NPF process in

upper troposphere (Zhao et al., 2020; Curtius et al., 2024). The observed ultrafine-particle-rich layer in Amazon's upper troposphere is mostly above 8 km, which is near the upper limit of the flight altitude range in this study. The Magdalena Mountains may act as a source of biogenic VOCs from local forests, as documented in previous field studies (e.g. Villanueva-Fierro et al., 2004). During the campaign, the large numbers of ultrafine particles at high altitudes were mainly observed on days with relatively strong convection (from July 25, flight C303 onwards). The observations indicate that NPF may be

common at high altitudes in this deep-convective environment, providing an important aerosol reservoir. This aerosol reservoir at high altitudes is significant for the climate system, as the small particles may grow to CCN size locally and impact local anvil and cirrus cloud properties. Possibly more importantly, as the newly formed particles grow with condensation and coagulation processes in subsiding cloud-free air, they may provide an important source of CCN to the lower troposphere and affect low-level cloud properties on regional scales (Williamson et al., 2019).

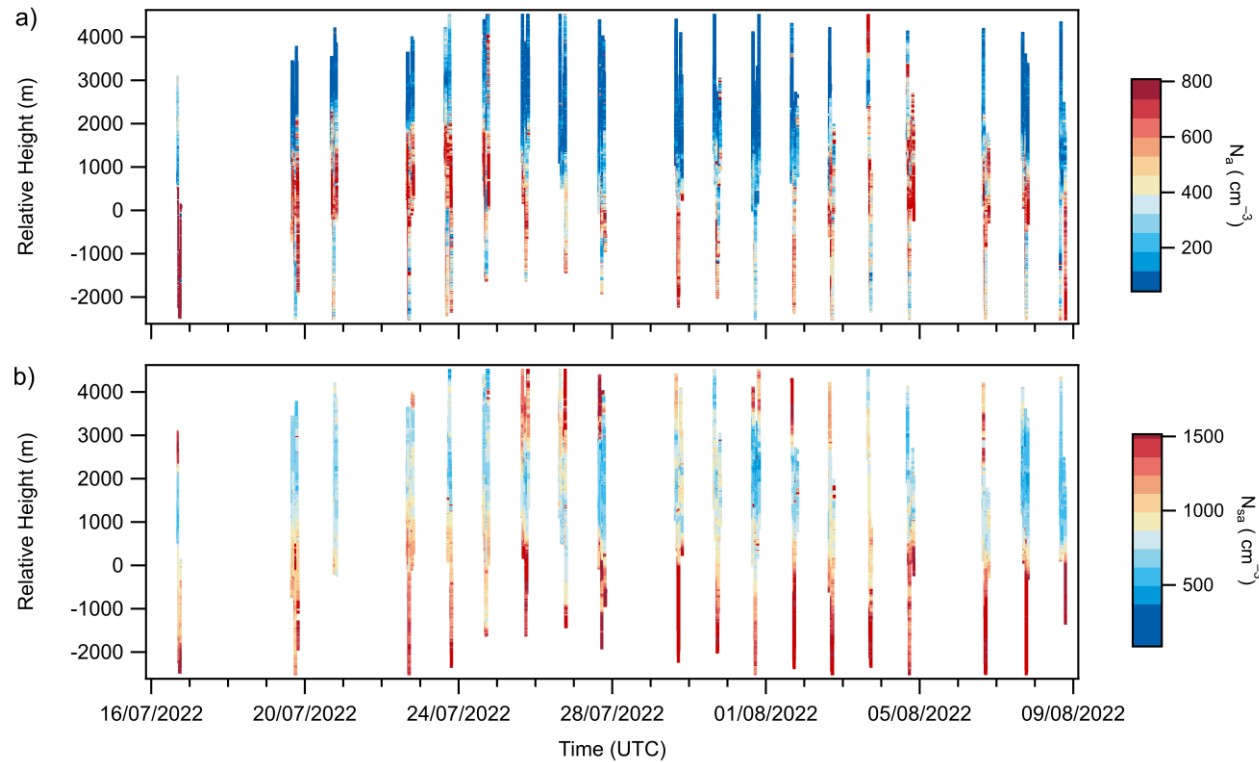

**Figure 3. Vertical profiles of a) out-of-cloud $N_a$ (0.1 to 3 μm) number concentrations from PCASP measurements and b) out-of-cloud $N_{sa}$ (2.5 nm to 1 μm) number concentrations from CPC measurements, for each flight. The y-axis is the relative height with respect to the LCL height.**

### 3.2.2 Aerosol physicochemical properties

As discussed in previous sections, the airmass sources, meteorological conditions, and vertical aerosol structures in the sampling area changed during the campaign. Here, available measured and estimated aerosol properties, including aerosol size distribution, chemical composition and hygroscopic characteristics, are summarized for: 1) Period 1 (available flights include C298 to C301, conducted from July 19 to 23), representing aerosol properties during the NW-flow (continental flow) period with relatively weak convection; and 2) Period 2 (available flights include C303 to C309, and C313, conducted from July 25 to August 1 and on August 6), representing aerosol properties during the SE-flow (oceanic flow) period with relatively strong convection.

*Size distribution*

Figure 4 shows number size distributions (dN/dlog$_{10}$Dp vs. Dp) and volume size distributions (dV/dlog$_{10}$Dp vs. Dp) of aerosols measured over near cloud-base SLRs for two periods. The analysis combined measurements from the SMPS and PCASP, covering a size range from about 20 nm to 3 μm. The corresponding average aerosol number size distributions for each flight are provided in Fig. S3. During Period 1, aerosol number size distributions for each flight followed a similar pattern

in terms of shape and concentration. These distributions were generally characterized by a weakly bimodal structure. The average number size distribution of Period 1 (Fig. 4a) had an Aitken mode with a peak concentration of ~800 cm$^{-3}$ at a diameter near 50 nm and an accumulation mode with a peak concentration of ~2600 cm$^{-3}$ at a diameter near 100 nm. During Period 2, aerosol number size distributions were generally more distinctly bimodal and showed larger variations than those during Period 1 (Fig. S3). Notably, in some flights, e.g. C303, the bimodal size distributions had an Aitken mode and an accumulation mode, with a distinct minimum between the two modes, commonly known as the Hoppel minimum (Hoppel et al., 1985). Compared to Period 1, the average number size distribution of Period 2 (Fig. 4b) presented a more pronounced Aitken mode, with a higher peak number concentration of ~2000 cm$^{-3}$ at a diameter similarly near 50 nm. In contrast, the accumulation mode was relatively reduced, showing a lower peak concentration of ~2000 cm$^{-3}$ at a slightly larger diameter near 115 nm. Though there were differences in number size distributions, the volume size distributions for Periods 1 and 2 were similar, as the aerosols at small sizes (< 0.1 μm) contributed only a small fraction to the total aerosol volume.

The more distinct bimodality observed during Period 2, particularly the Hoppel minimum observed in some flights, suggests that aerosols during Period 2 may have experienced greater cloud processing than those during Period 1. Larger particles are more likely to be activated as cloud droplets, which then either precipitate (in-cloud scavenging) or evaporate to form even larger aerosol particles due to in-cloud chemical processing and coagulation resulting from collisions between cloud droplets. Previous aircraft campaigns over the Amazon Basin reported that the detrained accumulation aerosols from precipitating clouds, which evaporate at the lateral boundaries, have relatively larger sizes but lower concentrations due to cloud-processing and scavenging (Braga et al., 2022). These processes could lead to the accumulation mode with a larger peak diameter and generally lower peak concentration observed during Period 2. The extent of precipitation may also contribute to the variations in the concentration levels of size distributions between individual flights during Period 2. The smaller (Aitken-mode) particles are typically formed from gas-to-particle conversion involving vapors like organic and sulfate vapors. The observed increase in Aitken-mode particle concentrations during Period 2 suggests that changes in source regions and meteorological conditions may have influenced the local gas-to-particle conversion process, as it is sensitive to environmental parameters such as T and RH (Kulmala et al., 2013). The lower surface T during Period 2 likely enhanced gas-particle partitioning by reducing the volatility of semi-volatile compounds, and the more humid boundary layer conditions likely further accelerated growth through the condensation of water-soluble gases.

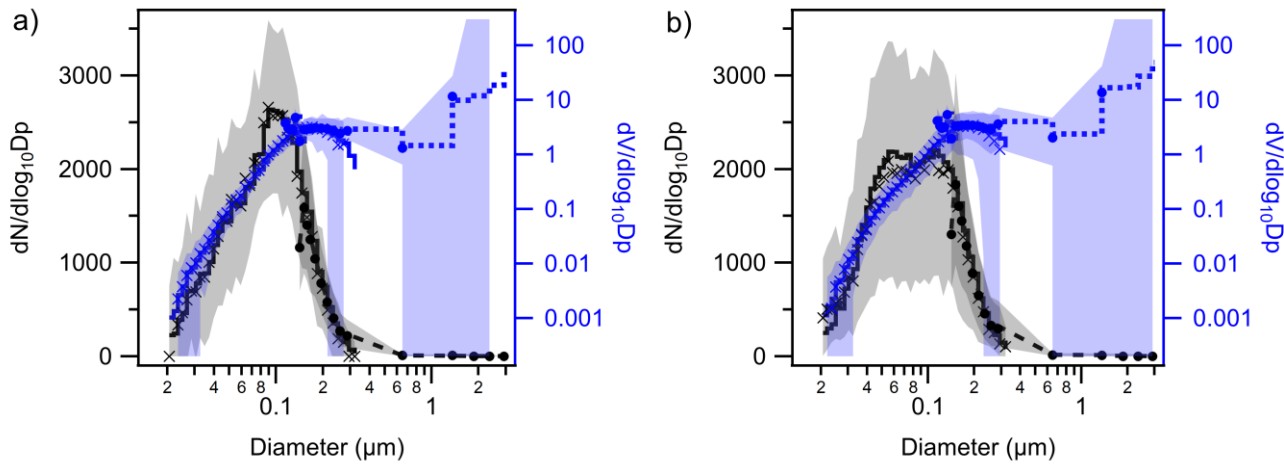

**Figure 4. Aerosol size distributions over near cloud-base SLRs for a) Period 1 and b) Period 2. The black lines, markers, and shades show average, median, 10 and 90 percentiles of number size distributions (left axis). The blue lines, markers, and shades show average, median, 10 and 90 percentiles of volume size distributions (right axis). The solid lines and cross markers represent results from SMPS; dashed lines and circle markers represent results from PCASP.**

*Chemical composition*

The average out-of-cloud vertical profiles of mass fractions of different aerosol compositions in the submicron-size range (PM1), measured by the AMS and SP2, are shown in Figs. 5a and 5b for Periods 1 and 2 separately. The corresponding absolute mass concentrations of different aerosol compositions for the two periods are provided in Fig. S4. During Period 1, the dominant PM1 component was organic throughout the aerosol column, followed by inorganic sulfate, ammonium, and nitrate. BC contributed a minor fraction to the total PM1, suggesting a potentially negligible influence of combustion emissions on local aerosol burdens during the campaign. There were variations in chemical composition below and above the LCL, in terms of concentrations and fractional contributions. This variation in aerosol layers may be partly due to slight differences in airmass source contributions at different altitudes, which is supported by NAME backward-dispersion results in Sect. 3.1. Other processes, such as in-cloud scavenging and chemical thermodynamics, may also contribute to this variation. Both absolute and relative sulfate levels were generally lower above the LCL compared to below, as the hygroscopic sulfate aerosols could act as efficient CCN and undergo in-cloud scavenging at and above the cloud base. It is noted that both absolute and relative nitrate levels generally increased with altitude, despite its hygroscopic nature. Inorganic nitrate, in the form of ammonium nitrate ($NH_4NO_3$), is a semi-volatile species. The colder and moister atmosphere at higher altitudes may promote the condensation of nitrate and ammonia vapors ($HNO_3 + NH_3$) onto existing aerosol particles, which has been reported in previous observational studies (Morgan et al., 2020; Wu et al., 2020). Therefore, the chemical thermodynamics of the $HNO_3$-$NH_3$-$NH_4NO_3$ system across large temperature and RH gradients (Fig. S5) may result in the condensation and formation of $NH_4NO_3$ above the LCL in this study.

During Period 2, mass concentrations of different aerosol compositions substantially decreased above the LCL, reaching minimal levels at ~1 km above the LCL, which followed the observed $N_a$ pattern. Compared to Period 1, the distinct chemical

change during Period 2 was the enhanced inorganic sulfate levels, in terms of absolute mass concentrations below the LCL and its fractional contribution throughout the aerosol column. This change could be due to the large intrusion of marine airmasses from the Gulf of Mexico during Period 2. Marine emissions of dimethyl sulfide (DMS) are the most important biological precursors of global sulfate aerosol, the oxidation of which forms sulfur dioxide ($SO_2$) and methyl sulfonic acid (MSA) (Hoffmann et al., 2016; Tashmim et al., 2024). $SO_2$ can be further oxidized by the gas-phase reaction with hydroxyl radical (OH) to form sulfuric acid ($H_2SO_4$), which is a significant contribution to aerosol nucleation and condensational growth. $SO_2$ oxidation through aqueous chemistry in cloud droplets, such as the reaction with hydrogen peroxide ($H_2O_2$), can also contribute to $SO_4^{2-}$ formation. In addition, previous studies reported a relatively higher emission density of anthropogenic $SO_2$ and subsequent near-surface sulfate concentrations in the eastern U.S. compared to the western U.S., largely due to emissions from power plants and industrial activities (Yang et al., 2017; Zhong et al., 2020). Therefore, both the marine emissions from the Gulf of Mexico and the large anthropogenic sulfur emission in the eastern U.S. likely contributed to the increase in sulfate levels in Period 2 under SE flow-controlled condition. Figure 5c shows the vertical profiles of O:C ratios for Periods 1 and 2, which is a proxy for the oxidation state of organic matter (Aiken et al., 2008). Another distinct chemical change during Period 2 was the higher O:C ratios compared to Period 1, suggesting a more oxidized organic state or/and a larger fraction of highly oxidized organics. OH radical is the dominant oxidizing agent in the troposphere. Model and satellite studies of global OH distributions have reported more abundant OH at lower latitudes (0–30° N) compared to higher latitudes (Zhao et al., 2019; Pimlott et al., 2022). The organics and organic precursors (VOCs) during Period 2 likely underwent more extensive photo-oxidation while being transported with the SE flow, consequently resulting in the more oxidized organic state.

*Hygroscopic parameter*

The hygroscopicity of aerosol particles determines their CCN ability to be activated into the cloud droplet, which is related to aerosol chemical compositions and size (Farmer et al., 2015). Kappa ($\kappa$), a parameter to evaluate the hygroscopic property of aerosols (Petters and Kreidenweis, 2007), was estimated based on chemical compositions measured by the AMS and SP2 (Haslett et al., 2019). In brief, the $\kappa$ values and volume fractions of different chemical species (such as BC, organics, ammonium sulfate, ammonium nitrate, etc.) were combined to derive a bulk $\kappa$ for the mixed particles, following the ZSR (Zdanovskii, Stokes and Robinson) mixing rule (Petters and Kreidenweis, 2007). The molar concentrations of different inorganic salts (such as ammonium sulfate, ammonium nitrate, etc.) were established using the measured components (nitrate, sulfate, and ammonium fragments) from the AMS and the ion pairing scheme, as shown in Haslett et al. (2019). Figure 5d shows the vertical profiles of calculated bulk $\kappa$ values, suggesting generally more hygroscopic PM1 during Period 2 compared to Period 1. The enhanced hygroscopic property of aerosols during Period 2 is likely due to the increase in the fractions of non-sea-salt inorganics in PM1 (Figs. 5a and 5b), as inorganic salts typically have higher $\kappa$ and are more hygroscopic than organics on a global scale (Pöhlker et al., 2023). Additionally, previous studies indicate that the hygroscopic parameter for organics, $\kappa_{org}$, is positively correlated with the organic O:C ratio (Chang et al., 2010; Han et al., 2022). An empirical linear relationship between $\kappa_{org}$ and O:C ratio ($\kappa_{org} = (0.29 \pm 0.05) * O/C$) was applied in this study, which is based on previous field observations at a rural site subject to biogenic and anthropogenic influence (Chang et al., 2010). The enhanced organic

oxidation state during Period 2 may have contributed slightly to the increase in $\kappa_{org}$ and thus bulk $\kappa$. Aerosol observations indicate that the source airmass regions and transport pathways play an important role in modifying aerosol size distributions and chemical compositions, and subsequently affecting their hygroscopic characteristics in this area, suggesting a potential impact on local cloud formation.

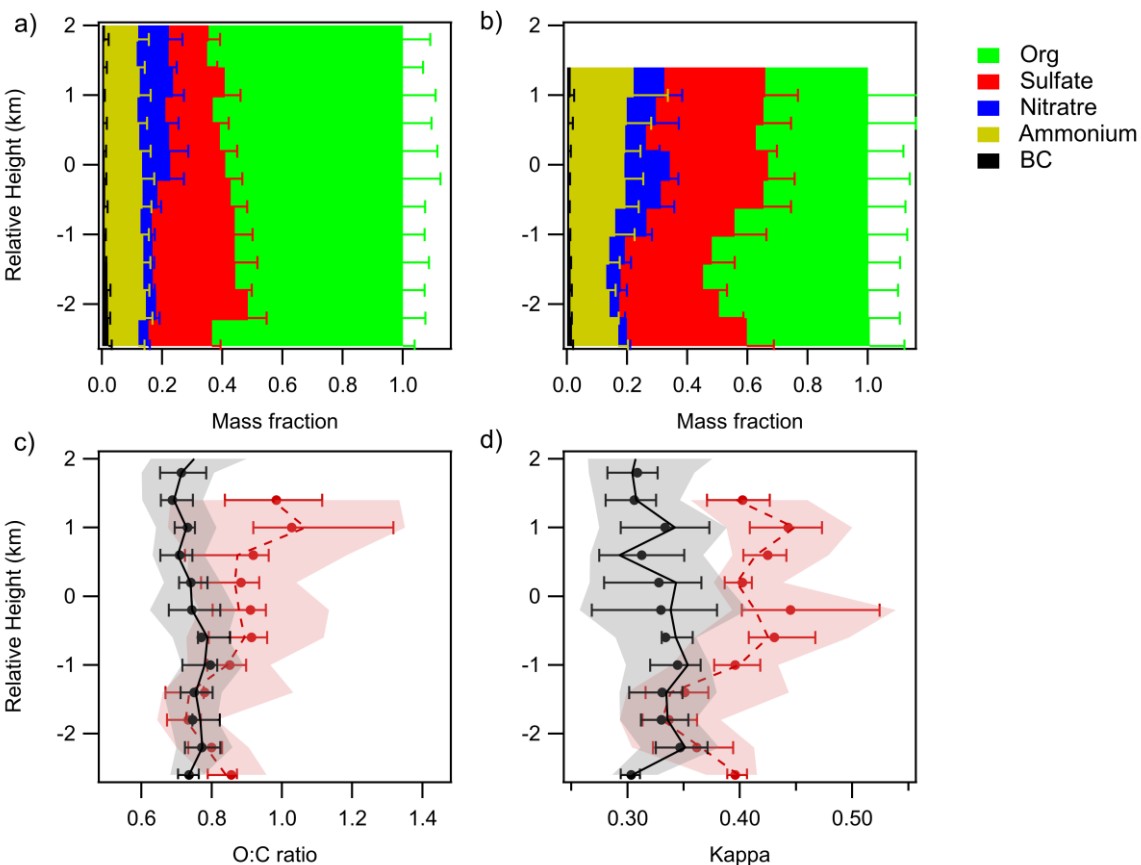

**Figure 5. Average vertical profiles of out-of-cloud PM1 chemical composition ratios for a) Period 1 and b) Period 2. The width of colour bars represents the average mass ratios of different species in each vertical bin. The error bars represent standard deviation. c, d) Vertical profiles of c) organic O:C ratio and d) bulk aerosol kappa for Period 1(black) and Period 2 (red). The lines, dots, bars and shades represent median, mean, 10, 25, 75, and 90 percentiles. The y-axis is the relative height with respect to the LCL height.**

### 3.3 Development of liquid clouds in the deep-convective system

### 3.3.1 Cloud droplet measurements

At the beginning of the campaign (flights C298-C300), the lowest measured cloud layers were at an altitude range of ~ (4.8–5.1) km a.s.l., with ambient T ranging between ~273 and 276 K. From July 23 (flight C301) onward, the lowest measured cloud layers were at a lower altitude range of ~ (3.8–4.4) km a.s.l., with ambient T ranging between ~ 275 and 280 K. Cloud properties in the lowest measured cloud layers during each flight are summarized in Table S2, along with the $N_a$ within 200 m

below the LCL height to represent close to cloud-base aerosol (0.1 to 3 µm) concentrations. Approximate distances between the lowest measured cloud layers and the LCL heights are also summarized in Table S2. Altitudes of the lowest measured cloud layers during each flight were generally close to or much higher than the estimated LCL height which approximates
cloud base height. It is noted that the actual cloud base could have been below the altitudes of the lowest measured cloud layers. Additionally, vertical profiles of measured cloud droplet properties were analysed to examine the vertical evolution of convective clouds during the campaign. Flights C297, C302, C311, and C315 are excluded from cloud analysis due to the absence of convective clouds or insufficient multi-altitude cloud sampling on those days.

Flights C298 to C301 (belong to Period 1) were characterized as the NW-flow dominated period, associated with relatively
weak convection and continentally influenced aerosol characteristics. Average values of below-cloud $N_a$ for these Period 1 flights were similar, in a range of $(533 - 636)$ cm$^{-3}$, with maximum values reaching $(1198 - 1490)$ cm$^{-3}$. The lowest measured cloud layers during these flights were consistently close to the estimated LCL heights (near the cloud base). Average values of $N_d$ in these lowest measured cloud layers near the cloud base were in a range of $(521 - 722)$ cm$^{-3}$, with maximum values reaching $(777 - 1055)$ cm$^{-3}$. Corresponding average values of LWC and $R_e$ ranged from 0.08 to 0.17 g m$^{-3}$ and 3.2 to 4.6 µm,
respectively. With similar LCL heights and atmospheric conditions, flights C298 to C300 had relatively consistent vertical development of cloud droplets. Figure 6 (black lines) shows the vertical profiles of $N_d$, LWC, and $R_e$ observed during flights C298 to C300. There was a slight decrease in $N_d$ at low cloud layers, followed by relatively constant values with height. The LWC and $R_e$ generally showed increasing trends with height, with some cases decreasing near the tops of the measured cloud layers. Observed maximum values of LWC and $R_e$ reached $3.0 - 3.5$ g m$^{-3}$ and $9.7 - 10.2$ µm, respectively, at approximately
3.3 to 3.8 km above the LCL where T ranged between 248 and 253 K. With a decreased LCL height, Flight C301 showed similar $N_d$ but slightly higher ranges of LWC and $R_e$ compared to earlier Period 1 flights (Fig. S6). Maximum LWC and $R_e$ values reached 5.8 g m$^{-3}$ and 12.2 µm, respectively near the top of the measured cloud layers, ~4.8 km above the LCL at the T of ~246 K.

Period 2 (flights C303 to C309, C313 and C314) was characterized as the SE-flow dominated period, associated with
480 additional changes in such as convection and aerosol properties. Compared to Period 1, Period 2 had a broader range of below-cloud $N_a$, with average values ranging from 298 to 795 cm$^{-3}$ and maximum values between 508 and 1202 cm$^{-3}$. During Period 2, the LCL heights were decreased compared to Period 1. Due to flight safety restrictions above the Magdalena Mountains, the lowest measured cloud layers were generally well above the cloud base, limiting direct comparisons of the lowest measured cloud layers between Periods 1 and 2. Nevertheless, in a subset of Period 2 flights (C305, C309, C313, and C314), the lowest
measured cloud layers were close to the estimated LCL heights (near the cloud base). These cases indicate that near cloud-base LWC (0.21 to 0.39 g m$^{-3}$) and $R_e$ (4.7 to 5.6 µm) during Period 2 were likely higher than those during Period 1. The vertical profiles of $N_d$, LWC, and $R_e$ during Period 2 (Fig. 6, red lines) followed similar patterns to Period 1. However, Period 2 flights generally had higher ranges of LWC and $R_e$ at equivalent heights above the LCL, compared to Period 1 flights. For example, in flights C303 to C306, the maximum LWC and $R_e$ values reached $4.6 - 5.9$ g m$^{-3}$ and $11.6 - 13.7$ µm, respectively,
near the tops of the measured cloud layers at T comparable to those in Period 1 $(< 253$ K). It is noted that cloud measurements

did not extend to the ambient cloud top in this study. Satellite estimates of the maximum cloud top heights are provided in Finney et al (2024), showing generally higher cloud tops during Period 2 (10.9 – 14.8 km a.s.l.) compared to Period 1 (7.6 – 12.7 km a.s.l.).

The decrease in LCL and associated cloud-base heights from July 23 (flight C301), along with higher cloud-base T and P, allowed for a larger amount of water vapour and subsequently led to the observed higher LWC occurred above the cloud base during Period 2 (Pruppacher and Klett, 2010). Compared to Period 1, Period 2 also exhibited significant changes in aerosol amounts and properties. During Period 2, aerosols close to cloud-base were generally more hygroscopic (average $\kappa$ = 0.34 – 0.53) than those during Period 1 (average $\kappa$ = 0.27 – 0.38), which was due to changes in aerosol composition associated with airmass source transfer. This suggests that aerosol particles during Period 2 were likely more effective at absorbing water

vapour and being activated into cloud droplets at the cloud base (Petters and Kreidenweis, 2007). The critical activation diameter could shift to a lower size during Period 2, driven by the presence of more hygroscopic aerosols. Another distinct change in aerosol properties, as discussed in Sect. 3.2, is the greater variability in aerosol concentrations and size distributions during Period 2. In some flights, the decrease in accumulation-mode particle concentrations may have counteracted the positive effect of enhanced hygroscopicity on aerosol activation and cloud droplet formation at the cloud base. However, increases in

Aitken-mode particle concentrations were observed in most of the Period 2 flights, which may have led to more small particles being activated to form additional cloud droplets at and above the cloud base, due to the likely smaller critical sizes for droplet formation during Period 2. A droplet number closure study by Braga et al. (2021) showed that Aitken mode particles, particularly those with high hygroscopicity, can play an important role in cloud-base droplet formation under conditions of low pollution levels and high updraft velocities in convective clouds. Observations and numerical simulations of deep

convective clouds over the Amazon also suggest that an increase in Aitken mode particles may also lead to the formation of additional cloud droplets above the cloud base (Fan et al., 2018). In summary, the observed changes in meteorological conditions (higher close to cloud-base T and P) and aerosol properties (varying aerosol concentration and size distributions, greater hygroscopicity) could influence cloud droplet formation and growth in this region, leading to higher ranges of LWC and $R_e$.

Differences in the formation and vertical evolution of cloud droplets may further influence rain/ice initiation and later precipitation processes in convective cloud systems. Previous studies have suggested that a range of $R_e$ between 12 and 14 μm indicates the height at which raindrops start to form via collision-coalescence (Rosenfeld et al., 2012, Braga et al., 2017b). Additionally, cloud droplet sizes ($R_e$) play a role in the formation and growth of ice particles. Previous satellite studies reported that glaciation temperatures of convective clouds were strongly dependent on $R_e$ at the −5 °C isotherm, with smaller $R_e$ delaying

glaciation (Rosenfeld et al., 2011). At the beginning of the campaign (C298-C300), observed $R_e$ values were mainly below 10 μm, suggesting suppressed conditions for cloud drop formation. Moreover, the presence of too small supercooled raindrops at cold temperatures likely inhibited ice production (i.e. riming and shattering) processes, while the precipitation may be overwhelmingly produced by ice processes in deep-convective systems (Korolev et al., 2017). This suppression of both raindrop and ice production was likely linked to the observed absence of precipitation during these days. From July 23 (flight

C301) onward, observed $R_e$ values were generally close to or exceeded 12 µm near the tops of measured growing cumulus clouds, suggesting conditions more favourable for raindrop and ice formation, and subsequent precipitation. This aligns with the observed increase in both the frequency and intensity of precipitation starting from July 23 (Fig. 2).

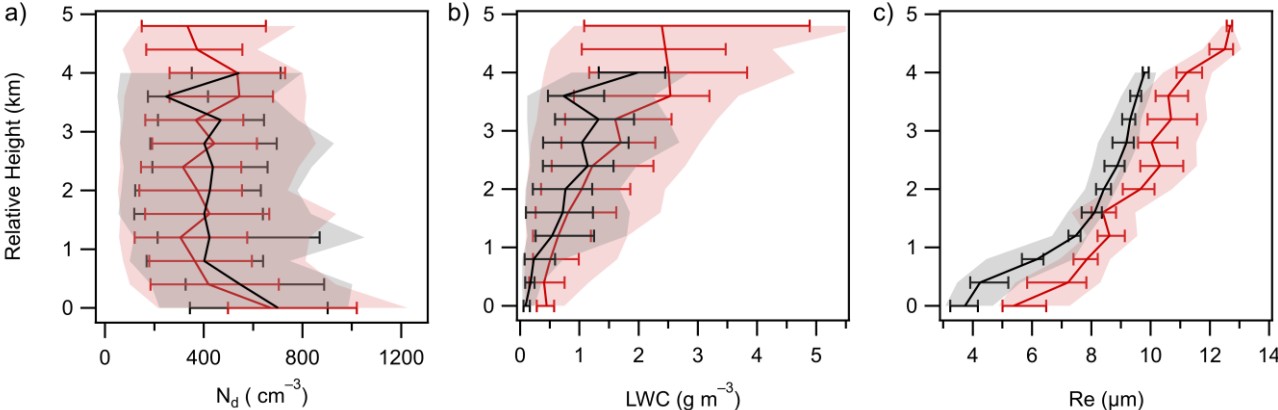

**Figure 6. Vertical profiles of a) cloud droplet number concentration ($N_d$, cm$^{-3}$), b) liquid water content (LWC, g m$^{-3}$), and c) effective**
**radius ($R_e$, µm) for Period 1 earlier flights (black, C298-C300) and Period 2 (red). The lines, bars, and shades represent median, 10, 25, 75, and 90 percentiles. The y-axis is the relative height with respect to the LCL height.**

### 3.3.2 Simulations from bin-microphysical parcel model

Simulations of cloud droplet formation and development were performed using the bin microphysics parcel model for two selected cases (C300 and C303, as highlighted in Fig. 1b), representing Periods 1 and 2, respectively. The initial conditions
for model simulations are summarized in Table S3 and Fig. S7, including such as the T, P, RH, aerosol $\kappa$ and size distributions close to cloud-base derived from airborne measurements. Figure 7 shows the vertical profiles of simulated $N_d$, LWC, and $R_e$ for the two cases, as well as their comparison with CDP measurements. Additionally, Fig. 8 shows the evolution of simulated and measured droplet size distributions for the two cases, as a function of T ranging from near cloud base to high level. Simulations assuming adiabatic conditions (black dashed lines in Fig. 7) yielded relatively constant $N_d$ with altitude, missing
the observed slightly decreased $N_d$ at low cloud layers. The simulated LWC and $R_e$ increased with height as did the observations. However, the adiabatic simulations largely overestimated $N_d$ and LWC. Though the simulated $R_e$ values were only slightly overestimated, the droplet size distributions showed significant differences between the simulations and observations (black dashed lines in Fig. 8). Observations of droplet size distributions typically presented wide ranges, with the width of size distributions increasing with height. In the Period 1 case (C300), CDP observations showed that most droplets near cloud-base
had a diameter smaller than 20 µm, which grew to larger sizes with vertical development of clouds. In contrast, the Period 2 case (C303) showed larger and wider size ranges as the cloud developed. However, the adiabatic simulations were much narrower than the observed distributions, which increasingly overestimated the cloud droplet size range as vertical development progressed. Previous adiabatic simulations of deep convective clouds over the Amazon have similarly demonstrated substantial discrepancies between predicted and measured $N_d$ at the cloud base, particularly in cases with high

updraft velocities ($> 2.5$ m s$^{-1}$) (Braga et al., 2021). Yum and Hudson (2005) also highlighted significant differences in the magnitudes of observed and simulated widths of the size distributions in shallow cumulus clouds from their adiabatic parcel model. These differences in the predicted and measured convective clouds are attributed to neglecting non-adiabatic processes and/or the entrainment of aerosol particles (Braga et al., 2021).

Many observations and modeling studies have suggested that entrainment and mixing processes are important for the
555 evolution of cloud microphysics (Burnet and Brenguier 2007; Lehmann et al., 2009; Braga et al., 2017a). In this study, three entrainment scenarios were examined, including purely homogeneous mixing (denoted as "Hom"), purely inhomogeneous mixing ("Inhom"), and a hybrid approach of early-stage inhomogeneous mixing (from the cloud base up to ~ 1 km above) followed by homogeneous mixing ("Inhom+Hom"). In the simulations, the entrainment rate of surrounding air was constrained by thermodynamic profiles derived from dropsonde measurements. The simulated entrainment rates under different
entrainment scenarios are shown in Fig. S8. With the inclusion of entrainment processes, which considered the dilution of cloud layers by entrained subsaturated air and the associated evaporation of cloud droplets, the simulated $N_d$ and LWC were substantially reduced throughout the cloud depth compared to adiabatic simulations. As seen in Figs. 7a and 7b, the inclusion of entrainment improved the agreement between simulated and observed LWC profiles (red, blue, and green solid lines). Simulated cloud-base $N_d$ values were more consistent with observations under the "Inhom" and "Inhom+Hom" scenarios (blue
and green solid lines in Figs. 7c and 7d), whereas they were overestimated under the "Hom" scenario (red solid lines in Figs. 7c and 7d). However, all three scenarios exhibited a pronounced decrease in $N_d$ with vertical development, in contrast to the relatively constant $N_d$ in the observations. Simulated $R_e$ values were substantially overestimated under the "Inhom" and "Inhom+Hom" scenarios (blue and green solid lines in Figs. 7e and 7f), due to the underestimated $N_d$, particularly at higher cloud levels. The "Hom" scenario produced $R_e$ profiles closer to observations (red solid lines in Figs. 7e and 7f) but continued
to predict excessively narrow droplet size distributions (red solid lines in Fig. 8). All three entrainment scenarios failed to reproduce small cloud droplets as vertical development progressed (red, blue, and green solid lines in Fig. 8).

Previous studies suggest that aerosols entrained with surrounding subsaturated air can be activated in the rising air parcel, broadening droplet size distributions toward small droplet diameters (Lehmann et al., 2009; Chandrakar et al., 2016). To account for this effect, aerosol entrainment (EA) was further incorporated into the simulations under three entrainment
scenarios, denoted as "Hom+EA," "Inhom+EA," and "Inhom+Hom+EA" (red, blue, and green dashed lines, respectively in Figs. 7 and 8). The entrained aerosols were assumed to follow the same size distributions as those measured near the cloud base (Fig. S7). For both cases, the further inclusion of aerosol entrainment made a negligible difference to the simulated LWC under three entrainment scenarios. In the Period 1 case (C300), incorporating aerosol entrainment improved the consistency of simulated $N_d$ profiles with observations, capturing the slight decrease at low cloud layers followed by relatively constant
values aloft. In particular, the "Inhom+EA" and "Inhom+Hom+EA" scenarios produced $N_d$ values close to observations, whereas the "Hom+EA" scenario still overestimated $N_d$. Aerosol entrainment also broadened the droplet size distributions under three entrainment scenarios, effectively filling the deficit in small-droplet populations. Under the "Hom+EA" scenario, the overestimated $N_d$ resulted in fewer large cloud droplets and slightly underestimated $R_e$, particularly at higher cloud levels.

In contrast, the "Inhom+EA" scenario produced a higher concentration of large droplets relative to observations, as vertical

development progressed. This is likely due to excessive collision-coalescence in the simulations, as inhomogeneous mixing promotes the evaporation of some droplets while leaving others unaffected, thereby enhancing droplet growth and shifting the droplet size distribution toward larger diameters (Pruppacher and Klett, 2010). These results suggest that homogeneous mixing should be incorporated as vertical development progresses. Overall, the "Inhom+Hom+EA" scenario provided the best agreement with observed $N_d$ and cloud droplet size. In the Period 2 case (C303), aerosol entrainment presented the same effects

on simulated LWC, $N_d$, and cloud droplet sizes as in the Period 1 case. However, the two cases exhibited slightly different sensitivities to aerosol entrainment under the "Inhom+Hom+EA" scenario. While the simulated $N_d$, $R_e$ and droplet size distributions showed good agreement with observations in the Period 1 case, the $R_e$ and large-size droplets were slightly underestimated in the Period 2 case due to the overestimated $N_d$ as vertical development progressed. This discrepancy is likely due to different vertical profiles of aerosol concentrations and size distributions between the two cases. In the Period 1 case,

$N_a$ and aerosol size distributions remained relatively constant from the cloud base to above (Figs. S9a and b). In the Period 2 case, $N_a$ decreased monotonically with height, and larger particles were less abundant above cloud base compared with measurements near cloud base (Figs. S9c and d). The assumption of constant aerosol size distributions during the entrainment process aligned well with the Period 1 case. However, it likely resulted in overestimated $N_d$ in the Period 2 case, due to excessive entrained aerosols in simulations, which in turn led to underestimated $R_e$. Employing vertically resolved aerosol size

distributions in the model would likely have little impact under Period 1 conditions but could improve the representation of droplet number and size evolution under Period 2 conditions. Additionally, this discrepancy is likely attributed to the parameterization of the initial radius of ascending cloud parcels in simulations (Table S3). The simulated entrainment rate is sensitive to the parcel radius, with smaller radius resulting in stronger entrainment and more rapid dilution. However, since the initial radius cannot be directly constrained by measurements, this also introduces uncertainty in simulations. In this study,

simulations of cloud droplet formation and development based on different assumptions highlight the importance of incorporating the entrainment of surrounding subsaturated air, particularly aerosol entrainment. The incorporation of aerosol entrainment is crucial in reproducing the width of droplet size distributions. Although our SMPS measurements did not provide continuous vertical profiles of aerosol size distributions due to limited time resolution, future studies should consider an approach of parameterizing fractional entrainment constrained by observed vertical profiles of aerosol number concentrations,

to further improve model performance.

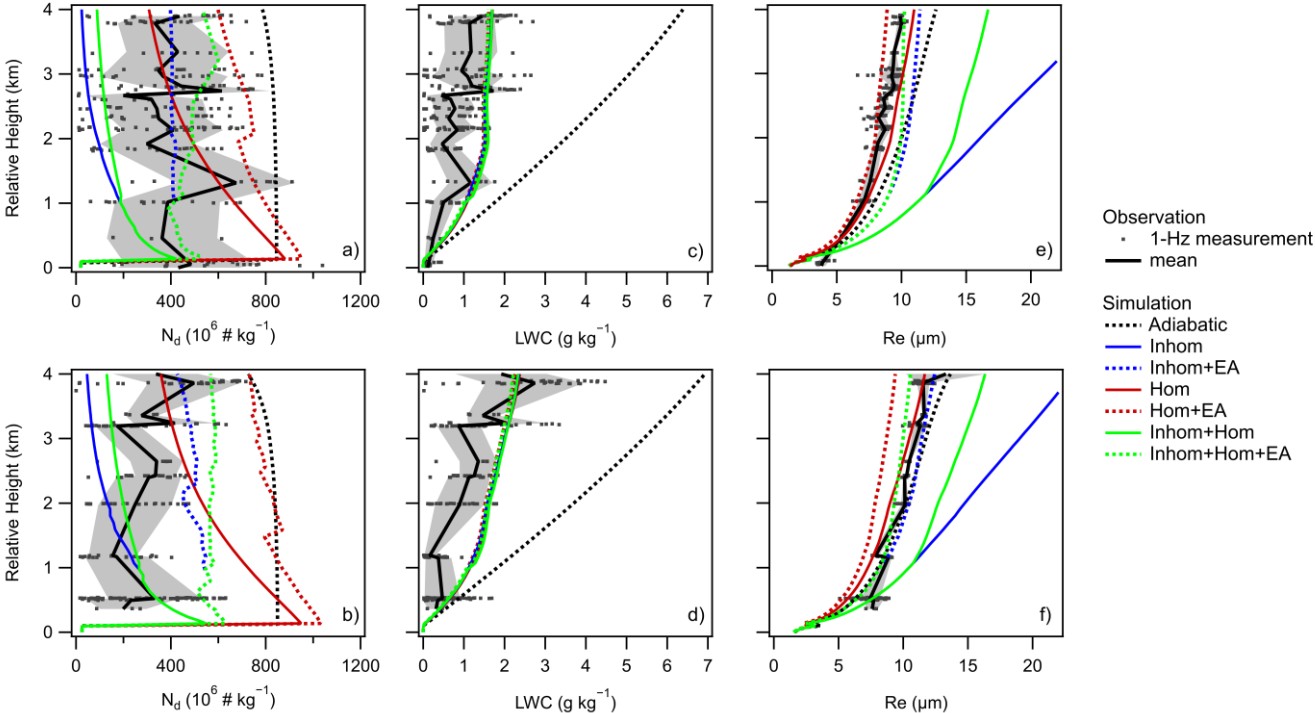

**Figure 7. Vertical profiles of observed and simulated cloud droplet number concentration ($N_d$, kg⁻¹), LWC (g kg⁻¹) and effective radius ($R_e$, μm) for the Period 1-C300 case (upper plots) and the Period 2-C303 case (bottom plots). It is noted that, for consistency with the model, the units of $N_d$ and LWC are expressed in "kg⁻¹", as opposed to "m⁻³" in Sect. 3.3.1. The dots represent 1-Hz measurements from the CDP. The black solid lines and shades represent averages and standard deviations of observed values. The black dashed, blue, red, and green lines represent simulations under different scenarios. The y-axis is the relative height with respect to the LCL height. It is noted that the pink and green dashed lines overlap in the plots of LWC.**

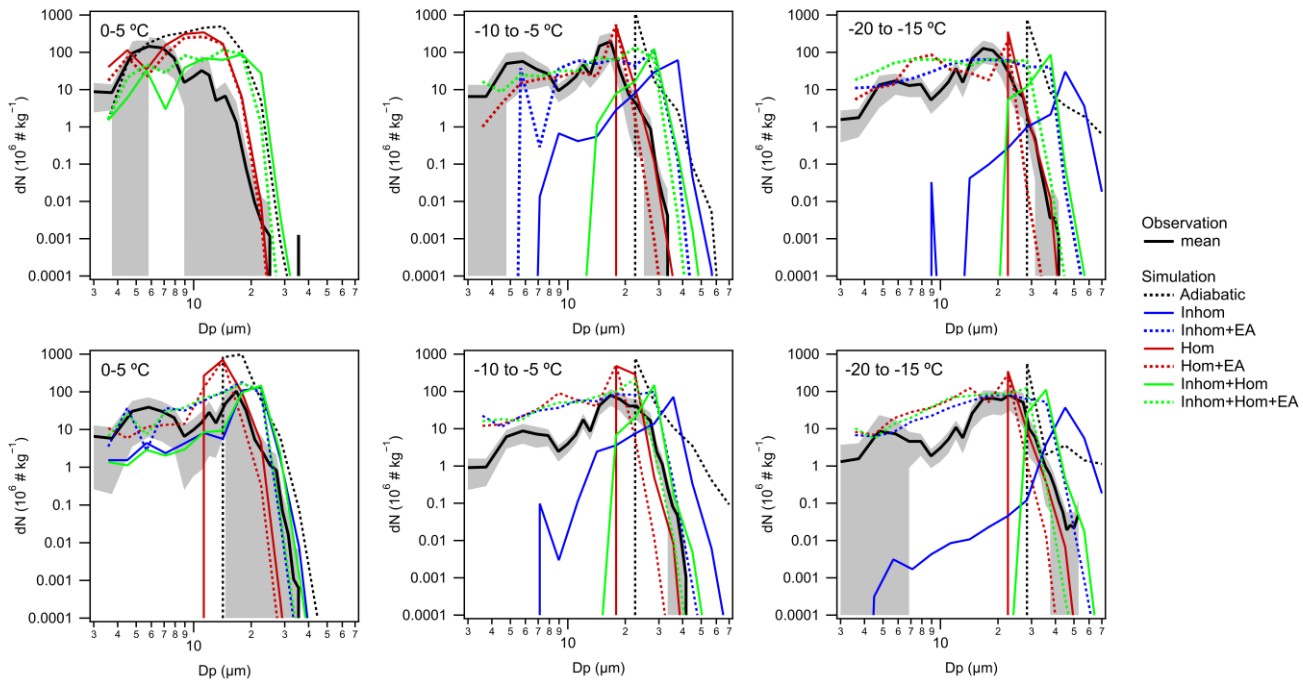

**Figure 8. Observed and simulated cloud droplet size distributions for the Period 1-C300 case (upper plots) and the Period 2-C303 case (bottom plots), as a function of T ranging from near cloud base to high level. The black solid lines and shades represent averages and standard deviations of observed values within specific T ranges. The black dashed, blue, red, and green lines represent simulations under different scenarios.**

## 4 Conclusions and implications

In this study, observations during the DCMEX campaign (July to August 2022) provide a comprehensive characterization of thermodynamics, aerosol and cloud droplet properties in deep convective systems over the Magdalena Mountains, New Mexico. These novel datasets provide valuable insights into the influence of meteorological conditions and aerosol characteristics on the formation and growth of cloud droplets in this and similar regions during summer season in North America.

Backward dispersion analysis of the sampling region was conducted using the UK Met Office's NAME. NAME results indicate that the airmass sources were initially dominated by NW continental flow at the beginning of the campaign, later transitioning to predominantly SE oceanic flow. Observations revealed that the changes in airmass sources were associated with changes in meteorological parameters. The SE-flow dominated period was generally characterized by lower LCL heights, stronger convection, higher surface humidity and lower surface T compared to the NW-flow dominated period. An increase in the frequency and intensity of precipitation was observed from July 23. Aerosol observations further highlight the influence of airmass sources and transport pathways on aerosol vertical distributions, size distributions, and chemical compositions, and subsequently their hygroscopic characteristics in this area. During the NW-flow period, $N_a$ was observed from the surface up

to altitudes above the LCL, while during the SE-flow period, $N_a$ was mainly concentrated within the LCL layer, potentially due to more efficient in-cloud scavenging processes. An enhancement in small particles at high altitudes was mainly observed on days with relatively strong convection, indicating NPF processes at high altitudes in this deep-convective environment.

Aerosol size distributions during the SE-flow period exhibited more pronounced bimodality compared to those observed during the NW-flow period. This was marked by an increase in Aitken-mode particle concentrations alongside generally fewer but larger accumulation-mode particles, suggesting greater cloud processing of aerosols under SE-flow conditions. The SE-flow period was also associated with larger sulfate mass fractions, more oxidized organic aerosol components, and subsequently greater hygroscopicity for submicron aerosols.

The observed changes in meteorological conditions and aerosol properties also influenced cloud droplet formation and growth, as indicated by CDP measurements. The SE-flow period generally presented higher ranges of LWC and $R_e$ compared to the NW-flow period. The differences were attributed to higher cloud-base T and P (warmer and lower cloud bases), as well as changing aerosol characteristics (greater variability in aerosol concentration, size distributions, and hygroscopicity) during the SE-flow period. The enhanced range of $R_e$ values (>12 µm) during the SE-flow period is likely correlated with the observed

increase in precipitation, underling the critical role of both dynamic and microphysical factors in shaping cloud properties and precipitation processes under different flow regimes. The increased precipitation during the SE-flow period could also be attributed to elevated water vapor content at warmer and lower cloud bases, which promotes additional condensation. Furthermore, a bin-microphysics parcel model was employed to simulate the formation and development of liquid cloud droplets under two representative conditions (NW-flow and SE-flow dominated periods) in this deep-convective system, which

was constrained by meteorological and aerosol parameters derived from field measurements. Different scenarios were tested, including idealized adiabatic conditions and cases incorporating entrainment processes from the surrounding environment. The idealised adiabatic simulations produced cloud droplet size distributions that were too narrow and the vertical structure of $N_d$ and LWC did not replicate observations. The model matched well with observations when considering entrainment processes from the surrounding environment, especially with the addition of aerosol mixing. The incorporation of aerosol

mixing with entrainment processes is crucial in reproducing the width of droplet size distributions.

This study provides a valuable dataset for improving parameterizations of aerosol-cloud interactions and offers critical constraints for the accurate representation of microphysical processes in this area and similar deep convective systems for future model studies. The comparison between observations and bin-microphysics parcel model simulations also highlights the importance of incorporating entrainment processes from surrounding environments, particularly aerosol mixing, in future

simulations of cloud droplets in deep convective systems. Future simulation studies should be conducted for all flight cases, to develop general parameterizations that achieve improved agreement between models and observations. In addition, future studies should investigate how different entrainment conditions (adiabatic, homogeneous, and inhomogeneous) affect not only the development of cloud droplets but also ice production.

*Data availability*. Airborne measurements are available from the Centre for Environmental Data Analysis https://catalogue.ceda.ac.uk/uuid/b1211ad185e24b488d41dd98f957506c/.

*Author contributions*. H.C., M.G., and A.B. designed the research; H.W., N.M., M.F., P.W., K.H., D.F., G.N., N.T., K.B., H.C., M.G., and A.B. performed field experiments; H.W., N.M., G.N., and D.F. prepared datasets; H.W. and P.C. performed simulations using a bin microphysics parcel model; H.W. performed NAME analysis; H.W. analyzed combined datasets and

led the manuscript writing.

*Competing interests*. The authors declare no competing interests.

*Acknowledgements*. This research has been supported by the Natural Environment Research Council (grant nos. NE/T006420/1 and NE/T006439/1). The authors (H.W. and P.C.) have received funding from Horizon Europe programme under Grant Agreement No. 101137680 via project CERTAINTY (Cloud-aERosol inTeractions & their impActs IN The earth sYstem).

Airborne data was obtained using the FAAM Airborne Laboratory BAe-146 Atmospheric Research Aircraft [ARA], operated by Airtask Ltd, managed by the National Centre for Atmospheric Science, leased through University of Leeds, and owned by UK Research and Innovation and the Natural Environment Research Council. The staff of Airtask, Avalon Engineering and FAAM are thanked for their professional work, before, during and after the deployment. The NAME group in the UK Met Office are thanked for their instructions on backward simulations.

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
