# Peer review of "The role of aerosols and meteorological conditions in shaping cloud droplet development in New Mexico summer deep-convective systems"

_EGUsphere, 2025_

## Author Comment (AC1)

**Reply to the Reviewer of Manuscript EGUSphere-2025-2600:**

We would like to sincerely thank the editor and reviewer for their time, effort, and thoughtful feedback on our manuscript. The reviewer comments are shown in **blue**, with the authors' responses shown in **black** and any edited manuscript language shown in *italicized black font*.

**General Comments**

This study beautifully documents the deep convective clouds that occur almost daily during summer over the Magdalena Mountains, which serve as a natural laboratory for continental convection. I enjoyed reading the detailed descriptions, which provide a comprehensive account and interpretation of the observations surrounding and within these clouds. The study describes the differences and probable causes of the aerosols, thermodynamics, and composition of the clouds under different air mass origins. These clouds, unsurprisingly, have high and cold bases, microphysically highly continental, with little or no significant warm rain processes. It follows that the clouds remain supercooled at least up to the -20 °C isotherm, which was the top of the measured flight levels. Apparently, only growing cloud towers were penetrated, because maturing clouds do glaciate at least occasionally at these high supercooled temperatures as they mature. So, please clarify the selection criteria for cloud penetrations.

We thank the reviewer for highlighting this important point.

The aircraft penetrations were not random but specifically guided by flight planning and operational constraints. Actively growing convective towers were preferentially sampled using real-time data and communication from the cloud radar network operators. In situ aircraft video was employed to provide an aircraft discussion to identify potential developing towers for sampling. These provided the best opportunity to capture supercooled liquid water and the early-stage microphysical evolution and ice processes. In contrast, mature or dissipating clouds were generally avoided or departed from, because of their lower likelihood of containing significant supercooled water, and stronger turbulence hazards for safe aircraft operations. Outflow regions following convective development were also identified for sampling when possible. Therefore, our dataset primarily represents developing cloud stages, which explains why full glaciation was rarely observed. In summary, the sampling strategy focused on early to middevelopment stages of convective clouds, where cloud tops were supercooled to at least –20 °C. This sampling strategy explains why complete glaciation was generally not encountered within the penetrated clouds, even though such processes could occur in maturing convective elements in the region.

The authors have added the related description in the method section:

"The sampling strategy focused on early to mid-development stages of convective clouds with supercooled tops. Mature or dissipating clouds were generally avoided or departed from, because of their lower likelihood of containing significant supercooled water, and stronger turbulence hazards for safe aircraft operations. As a result, complete glaciation was rarely encountered within the penetrated clouds, even though it occurred in maturing convective elements in the region."

The most enlightening part of the paper was the comparison of the clouds' vertical microphysical profiles with the parcel model, considering various assumptions. It showed the potential role of mixing in cloud drop activation and evaporation aloft. While informative, the paper lacks a scientific focus and a statement of novelty, i.e., where does it contribute fundamental understanding to the state of the art? This is evident in the fact that much of the introduction is devoted to issues not addressed by the findings of this study, such as the extensive description of the aerosol convective invigoration hypothesis.

This shortcoming can be overcome by focusing on the processes of cloud mixing (homogeneous vs. inhomogeneous) and the additional activation of drops versus evaporation aloft. To do that, I suggest:

- 1. Replace much of the irrelevant parts of the introduction with a review of the known background on the processes that cause deviations from adiabatic parcels in deep non-precipitating water clouds.
- 2. Review causes for shaping the cloud drop size distributions with height in such clouds.

Thanks to the reviewer for raising these insightful points.

**Regarding suggestions-1&2:** We have re-organized the introduction:**

- 1) The introduction of invigoration hypothesis has been deleted.
- 2) We now emphasize that detailed characterization of aerosol amounts and properties is crucial for understanding the droplet formation and development in deep convective systems. Moreover, atmospheric sources and transport generate temporally varying aerosol types, amounts, and properties in a given region.
- 3) We have added the background on how entrainment processes and aerosol mixing affect the development of convective clouds.

The revised introduction is:

"The life cycle of deep convective clouds is modulated by complex microphysical processes, including droplet formation, droplet growth through condensation and coalescence, thermodynamic phase transitions between liquid droplets and ice crystals, and the development of precipitation (Arakawa, 2004). Aerosols play a key role in these processes, and aerosol-cloud interactions are considered among the largest uncertainties in estimating climate sensitivity to radiative forcing (Boucher et al., 2013; Forster et al., 2021). Aerosols can affect clouds by acting as cloud condensation nuclei (CCN) or ice nucleating particles (INP), which is termed aerosol indirect effects (Boucher et al., 2013). In the presence of aerosols or sufficient CCN, water vapor condenses onto CCN surfaces to form cloud droplets, marking the initial process in the lifecycle of convective clouds that mostly occurs at cloud base (Tao et al., 2012; Rosenfeld et al., 2014). Generally, higher CCN concentrations produce a greater number of smaller droplets and narrower droplet size distributions, which are likely to inhibit collision-coalescence and delay raindrop formation, thereby extending cloud lifetime (Rosenfeld, 2000; Tao et al., 2012). This delay can have opposing effects on convective cloud development: the increased condensational heat release tends to enhance cloud buoyancy and vertical development, while the resulting increase in condensate loadings partially offsets that buoyancy enhancement (Rosenfeld et al., 2008; Koren et al., 2014; Fan et al., 2018; Varble et al., 2023). However, the presence of giant CCN, such as coarse-mode sea salt aerosols, can produce initially large droplets and accelerate warm rain formation, thereby inhibiting the vertical development of convective clouds (Yin et al., 2024). These existing studies suggest that the CCN ability of aerosols determines the initial droplet number concentration and size distributions, thereby influencing subsequent cloud dynamics throughout convective cloud lifetime. Additionally, INPs can promote the heterogeneous freezing, and regulate ice crystal number concentrations during convective development (Tao et al., 2012). Detailed characterization of aerosol amounts and properties is therefore crucial for improving the representation of aerosol-cloud interactions in atmospheric models, in particular, aerosol size distribution and chemical composition which determine their CCN and INP ability (Petters and Kreidenweis, 2007). However, the representation of aerosol properties and associated indirect effects is complex and uncertain, as they are subject to atmospheric dynamic and thermodynamic conditions (Yang et al., 2020). Atmospheric transport generates temporally varying aerosol types, amounts, and properties in a given region (Yang et al., 2020). Moreover, cloud response to aerosol perturbations depends on environmental conditions, such as cloud-base temperature, updraft velocity, and humidity (Pruppacher and Klett, 2010). Previous studies have indicated that clouds with cool, high bases tend to exhibit little sensitivity in cloud-top height and precipitation to aerosol loadings, while clouds with warm, low bases display larger aerosol-induced changes (e.g. Li et al., 2011). Overall, it is critical to understand how varying aerosol properties influence convective cloud microphysics under different environmental conditions.

Another key uncertainty in understanding the development of convective clouds is the entrainment and mixing process. Theoretically, cloud-droplet growth in a closed (adiabatic) parcel leads to narrower size distributions as vertical development progresses, tending to suppress the onset of coalescence through differential gravitational sedimentation (Pruppacher and Klett, 2010). However, observations have revealed that cloud-droplet size distributions are relatively broader than those in ideal adiabatic parcels, and this result is usually attributed to a consequence of entrainment and mixing and secondary activation (Blyth, 1993; Chandrakar et al., 2016). Despite its recognized importance, the representation of the entrainment and mixing process in models remains uncertain. It is suggested that inhomogeneous mixing typically dominates when cloud droplets are small, as their evaporation rates significantly exceed the mixing rate of clouds with surrounding subsaturated air (Pruppacher and Klett, 2010). In this way, a subset of cloud droplets evaporates completely, leaving the others in the volume unchanged. When cloud droplets are larger and their evaporation rates are comparable to the mixing rate, homogeneous mixing dominates the system and the influence of inhomogeneous mixing weakens (Pruppacher and Klett, 2010). With homogeneous mixing, droplets evaporate by uniformly reducing their size across the population, leaving droplet number density largely unchanged except through simple dilution. While some studies suggest that entrainment and mixing in convective clouds are almost completely inhomogeneous (e.g. Burnet and Brenguier, 2007; Braga et al., 2017a), some other studies propose that inhomogeneous mixing may dominate early cloud development when droplets are small, and then homogeneous mixing may become more prevalent as convective clouds evolve (e.g. Lehmann et al., 2009). It is unclear whether the entrainment is predominantly homogeneous mixing, inhomogeneous mixing, or a combination of both. An improved understanding of how these mixing mechanisms dominate throughout the development of convective clouds is essential for improving the representation of their microphysical processes. Moreover, the consequence of entrainment processes on cloud microphysical evolution can be regulated by surrounding environmental conditions, such as relative humidity (RH) and aerosol characteristics (Koren et al., 2010). In particular, aerosols entrained with surrounding air can act as additional CCN and promote secondary activation of droplets above cloud base. Such additional activation has been identified as an important factor contributing to the broadening of droplet size distributions, toward small droplet diameters (Lehmann et al., 2009; Chandrakar et al., 2016). An improved understanding of aerosol entrainment will reduce the current uncertainty in predicting droplet number concentrations and size distributions."

3. In the parcel simulations, provide the exact handling, formulation, and fraction of mixing with ambient air and secondary aerosol activation. Which extent of mixing would provide the best match with observations? Would replacing the aerosol size distribution near cloud base with its vertical profile improve the agreement between the simulated and observed cloud microstructure vertical profile?

**Regarding suggestion-3, we have revised the manuscript in four aspects as below:**

1) The authors have added the exact handling and formulation of mixing with ambient air and secondary aerosol activation in the supplementary. The added supplementary is:

**"S1 Representation of entrainment**

In this study, two types of entrainment processes are considered: homogeneous and inhomogeneous mixing. For homogeneous mixing, entrained subsaturated air uniformly mixes with and dilutes the cloud parcel, leading to a uniform reduction in droplet size. When humidity decreases sufficiently to induce droplet evaporation, the released aerosol particles may reactivate. These processes are implemented by including additional formulations in the solver routine that are passed to variable-coefficient ordinary differential equation (VODE) solver. The related formulations are described in Pruppacher and Klett (2010, chapter 12). In brief, we assume a "jet" parcel, which allows entrainment to occur through the front interface of the plume. For a jet parcel with mass (m), density ( $\rho$ ), radius ( $R_i$ )

and vertical velocity (W), entrainment is described in terms of a change in mass flux  $F_m = \pi R_j^2 \rho W$  along the vertical plume axis. The change in mass flux over a vertical distance  $\Delta z$  is expressed as  $\Delta z (dF_m/dz)$ , from which  $dF_m = 2\pi R_i \rho W dR_i$ . The entrainment rate for a jet  $(\mu_i)$  is therefore

$$\mu_j = \frac{1}{F_m} \frac{\mathrm{d}F_m}{\mathrm{d}z} = \frac{C}{R_j} \tag{S1}$$

where C is the important entrainment parameter, which is set as 0.2 based on previous laboratory studies. Entrainment will cause the parcel volume to increase with time. The parcel acceleration is calculated by considering the buoyancy and reaction of the surrounding air:

$$\frac{\mathrm{d}W}{\mathrm{d}t} = \frac{\mathrm{g}}{1+\gamma} \left( \frac{T-T'}{T'} - w_L \right) - \frac{\mu_j}{1+\gamma} W^2 \tag{S2}$$

where  $\gamma \approx 0.5$ , g is the acceleration due to gravity ( $g \approx 9.80665$  m s-2),  $w_L$  is the liquid water mixing ratio, T is the temperature of air parcel, and T' is the ambient temperature of the surrounding air. For a jet parcel, we can also relate the growth of the radius to the entrainment rate  $\mu_i$  according to

$$\frac{\mathrm{dln}R_j}{\mathrm{d}t} = \frac{1}{2} \left[ \mu_j W - \frac{\mathrm{dln}\rho}{\mathrm{d}t} - \frac{\mathrm{dln}W}{\mathrm{d}t} \right] \tag{S3}$$

which represents parcel dilution, expansion, and the conservation of mass flux, with  $\frac{d\rho}{dt}$  expressed in terms of pressure and temperature derivatives using the ideal gas law. The condensed water is related to the water vapour mixing ratio  $(w_v)$ , through an obvious statement of water conservation, this leads to

$$\frac{dw_v}{dt} = -\frac{dw_L}{dt} - \mu_j(w_v - w_v' + w_L)W \tag{S4}$$

Where  $w_L$  is the liquid water mixing ratio and  $w_v'$  is the ambient water vapour mixing ratio of the surrounding air. For the settings of aerosols, we initially set a number size distribution of dry aerosol particles  $(n_{AP,a}^{'})$  with mass  $(m_{AP}^{'})$ , where  $n_{AP,a}(m_{AP})$  is the number distribution of inactivated drops inside the air parcel at a time t. The  $n_{AP,a}(m_{AP})$  changes are due to (1) entrainment of additional aerosol particles from the environmental air, (2) the activation of some of the aerosol particles to drops, and (3) drops which by evaporation become deactivated particles again. These changes are considered in simulations, which are expressed by

$$\frac{\partial n_{AP,a}(m_{AP})}{\partial t} = -\mu_{j}W[n_{AP,a}(m_{AP}) - n'_{AP,a}(m_{AP})] + \frac{\partial n_{AP,a}(m_{AP})}{\partial t}\bigg|_{\text{activated/deactivated}}$$
(S5)

For inhomogeneous mixing, entrained subsaturated air remains in localized pockets rather than mixing instantaneously, leading to the evaporation of some droplets while others out of the pockets are unaffected. These processes are carried out outside the VODE solver, over a longer 10s timestep. The related formulations are the same as in the homogeneous mixing case (Eqs. S1-S5). For inhomogeneous mixing, the droplet number concentrations in each size bin are adjusted to conserve the parcel humidity, preventing a uniform reduction in droplet size as in the homogeneous mixing. When inhomogeneous mixing causes droplet evaporation, aerosol particles are released back into the discrete packets of subsaturated air, within the parcel, which may become re-activated later."

2) The mixing with ambient air is indicated by the strength of simulated entrainment rate  $(\mu_j)$ . The vertical profiles of simulated  $\mu_i$  are now shown in Fig. S8.

Figure S8. Vertical profiles of simulated  $\mu_j$  (m-1) for the Period 1-C300 case (left) and the Period 2-C303 case (right). The y-axis is the relative height with respect to the LCL height.

3) As described in the original manuscript, it is unclear whether mixing is predominantly homogeneous, inhomogeneous, or a combination of both. While some studies suggest that entrainment and mixing in convective clouds are almost completely inhomogeneous (e.g. Burnet and Brenguier, 2007; Braga et al., 2017a), some other studies propose that inhomogeneous mixing may dominate early cloud development when droplets are small, and then homogeneous mixing may become more prevalent as convective clouds evolve (e.g. Lehmann et al., 2009). In the revised manuscript, we examined three entrainment scenarios, including purely homogeneous mixing, purely inhomogeneous mixing, and a hybrid approach of early-stage inhomogeneous mixing (from the cloud base up to ~ 1 km above) followed by homogeneous mixing. We also examined the effects of aerosol entrainment. By comparing different scenarios, a combination of early-stage inhomogeneous mixing and following homogeneous mixing as well as the inclusion of aerosol entrainment would provide the best match with observations. The revised manuscript is:

[revised manuscript text omitted]

Figure S9. Vertical profiles of out-of-cloud  $N_a$  for a) the Period 1 case (C300) and c) the Period 1 case (C303). The lines, dots and shades represent median, mean, 10, and 90 percentiles. The black and red horizontal lines represent straight-and-level runs near the cloud base and above the cloud, respectively. The y-axis is the relative height with respect to the LCL height. b, d) Average aerosol size distributions during straight-and-level runs near the cloud base (black) and above the cloud (red) for b) the Period 1 case (C300) and d) the Period 1 case (C303). Lines and shades represent means and standard deviations.

4. Provide the formulation of the mix between homogeneous and inhomogeneous mixing, and which fractionation provides the best match to observations.

Regarding suggestion-4: The formulations for homogeneous and inhomogeneous mixing have been described in our response to suggestion-3, which are now included in the supplementary. In the revised manuscript, we examined three entrainment scenarios, including purely homogeneous mixing (denoted as "Hom"), purely inhomogeneous mixing ("Inhom"), and a hybrid approach of early-stage inhomogeneous mixing (from the cloud base up to  $\sim 1$  km above) followed by homogeneous mixing ("Inhom+Hom"). Under the "Inhom+Hom" scenario, we did not implement homogeneous and inhomogeneous mixing simultaneously. The two types of mixing were treated sequentially, rather than as a combined process with prescribed fractional contributions. By comparing different scenarios, a combination of early-stage inhomogeneous mixing and following homogeneous mixing as well as the inclusion of aerosol entrainment would provide the best match with observations.

There is a wealth of data from the individual flights, warranting an additional study that focuses on this, aiming to find the parameterization that best fits the individual flights. It is likely beyond the scope of this paper, but at the very least, state that this is a potential future study when addressing the most general questions above.

Thanks to the reviewer for raising this insightful point. We agree that the extensive dataset obtained from individual flights offers valuable potential for developing general parameterizations that achieve improved agreement between simulations and observations. While this study focuses on case studies using a bin-microphysics parcel model to highlight the importance of incorporating aerosol entrainment, we are preparing a follow-up manuscript to conduct simulations for all flights during the DCMEX campaign. We will systematically investigate how different entrainment conditions (adiabatic, homogeneous, and inhomogeneous) affect not only cloud droplets but also secondary ice production.

We have expanded the discussion of implications; the revised conclusion is:

"Future simulation studies should be conducted for all flight cases, to develop general parameterizations that achieve improved agreement between models and observations. In addition, Future studies should investigate how different entrainment conditions (adiabatic, homogeneous, and inhomogeneous) affect not only the development of cloud droplets but also ice production."

**Minor Comments**

Line 395: It is much more likely that the SO2 sources at the southeast are from urban and industrial emissions, including the extensive oil fields and refineries.

We thank the reviewer for this insightful comment. We have added the possible influence of urban and industrial SO2 emissions in the southeast U.S. The rephrased manuscript is:

In addition, previous studies reported a relatively higher emission density of anthropogenic SO2 and subsequent nearsurface sulfate concentrations in the eastern U.S. compared to the western U.S., largely due to emissions from power plants and industrial activities (Yang et al., 2017; Zhong et al., 2020). Therefore, both the marine emissions from the Gulf of Mexico and the large anthropogenic sulfur emission in the eastern U.S. likely contributed to the increase in sulfate levels in Period 2 under SE flow-controlled condition.

Line 500: Replace "raindrops" with "cloud drops". Accepted

Line 522: All cloud drop size distributions had a local maximum concentration at  $6.5 \mu m$  and a local minimum at  $8 \mu m$ . It appears to be a problem of incorrect bin widths for the CDP, rather than a bimodal drop size distribution being the issue.

We thank the reviewer for this suggestion.

In the revised manuscript, we have done CDP calibrations following methods in previous FAAM studies (e.g. Barrett et al., 2022). The updated droplet size distributions (dN vs. Dp) continue to exhibit a generally bimodal structure (Fig. 8). We also plotted the CDP size distributions as dN/dlogDp vs. Dp (Fig. R1), which were normalized by bin width. Figure R1 also indicates a generally bimodal structure when bin-width normalization is applied.

However, to mitigate the concern proposed by the reviewer, we have rephrased the relevant sentence in the manuscript as follows:

"Observations of droplet size distributions typically presented wide ranges, with the width of size distributions increasing with height."

Figure R1. Observed cloud droplet size distributions for the Period 1-C300 case (left) and the Period 2-C303 case (right), as a function of T ranging from near cloud base to high level. The solid lines and shades represent the averages and standard deviations of observed values within specific T ranges.

**Line 555 and the whole paragraph: How was the mixing performed in the model? And how was the portion of homogeneous and inhomogeneous mixing determined?**

In the revised manuscript, we examined three entrainment scenarios, including purely homogeneous mixing (denoted as "Hom"), purely inhomogeneous mixing ("Inhom"), and a hybrid approach of early-stage inhomogeneous mixing followed by homogeneous mixing ("Inhom+Hom"). Under the "Inhom+Hom" scenario, we initiated the entrainment with inhomogeneous mixing from the cloud base up to approximately 1 km above it. Beyond this altitude, the simulations transitioned to homogeneous mixing. We did not implement homogeneous and inhomogeneous mixing simultaneously. The two types of mixing were treated sequentially under the "Inhom+Hom" scenario, rather than as a combined process with prescribed fractional contributions.

**Lines 613-614: The added precipitation with warmer bases can be explained by the increased water vapor content and the corresponding additional condensation. Please add this as a further possible explanation.**

We thank the reviewer for this suggestion. We have added the suggested explanation in the revised manuscript.

"The increased precipitation during the SE-flow period could also be attributed to elevated water vapor content at warmer and lower cloud bases, which promotes additional condensation."

**Fig. S1: Please state the heights of the origins of the back tracks.**

The release heights of the backward-dispersion simulations have been added to Fig. S1.

**References**

[revised manuscript text omitted]

---

## Author Comment (AC2)

**Reply to the Reviewer of Manuscript EGUSphere-2025-2600:**

We would like to sincerely thank the editor and reviewer for their time, effort, and thoughtful feedback on our manuscript. The reviewer comments are shown in **blue**, with the authors' responses shown in **black** and any edited manuscript language shown in **italicized black font**.

This manuscript deals with the relationship between aerosols, meteorological conditions and the development of cloud droplets and their size distribution. The methodology is based on both aircraft observations and numerical parcel model with bin microphysics. The overall goal is to understand the role of aerosols (number and type), constrained by meteorology, on the droplet formation mechanisms. The authors argue that aerosol entrainment is an important mechanism to explain the broadening of the droplet spectra. Overall, I think the manuscript is deserving of publication, but there are some major and minor adjustments to be made. I will point some major comments here and list a few minor comments later.

**Major Comments**

- 1. There is a disconnect between what the manuscript seems to be by reading the title, abstract and introduction and what the manuscript actually is. When I started to read the manuscript I thought it was going to be about the invigoration hypothesis, which is extensively reviewed in the introduction. However, the results section dedicates a lot of text to the description of the aerosol properties and air mass backtrajectories. While interesting, it doesn't seem to fit what I expected from the title/abstract/introduction. The same could be said about the entrainment process, which seems to be a major focus of the study and is not mentioned in the title, for instance. I will give my suggestions together with my second comment below.
- 2. The origin of the issue listed above is possibly related to the many foci present in the manuscript:
- I) There is discussion about air masses/backtrajectories
- II) There is discussion on physicochemical properties of aerosols, with vertical profiles
- III) Only then the manuscript goes into the discussion that I was expecting to see from the start, which is the droplet spectra properties
- IV) Compounding the many-foci issue, there is also a lot to cover from the observations side as well as from the modeling side. Therefore, there is also multiple foci on the methodology side.

With the given above, the manuscript ended up being, in my opinion, too long and without a clear message. I would suggest the authors rewrite the manuscript in a way to make their contribution more explicit. From reading the manuscript, I think the most interesting aspect was the comparison between the observations and the bin model with the entrainment discussion. Because the manuscript had so many foci, this discussion ended up being shorter and shallower than expected. For instance, the authors mention that the entrainment effect is important to reproduce the observed droplet spectra width. However, there is no figure showing this parameter explicitly. There is plenty of literature about the aerosol effect on droplet width, which could be used as context to the present study.

To summarize and provide a more objective suggestion:

I would suggest refocusing the title/abstract/introduction towards this entrainment effect and the observation-model comparison. The aerosol/backtrajectories analysis, while interesting, could be left out without hurting the overall message of the manuscript. This part could be later incorporated in a new submission focused on the aerosol discussion, in my opinion.

We sincerely thank the reviewer for these insightful points.

A primary objective of this manuscript is to provide a detailed characterization of aerosol properties and meteorological conditions during the DCMEX campaign, and how these factors influence the formation and development of cloud droplets in deep convective systems. Another publication (Daily et al., 2025) from the DCMEX project focuses on the characterization of ice-nucleating particles (INPs). The two studies establish the framework of aerosol characterization for DCMEX. In the revised manuscript, we therefore retain the discussion of aerosol physicochemical properties. Since the atmospheric sources and transport generate temporally varying aerosol properties in a given region, the source analysis using the dispersion model is needed to explain the observed temporal trends in aerosol characteristics. To address the reviewer's question about a disconnect between the introduction and results, we have reorganized the introduction section:

- 1) The introduction of invigoration hypothesis has been deleted.
- 2) We now emphasize that detailed characterization of aerosol amounts and properties is crucial for understanding the droplet formation and development in deep convective systems. Moreover, atmospheric sources and transport generate temporally varying aerosol types, amounts, and properties in a given region.
- 3) We have added the background on how entrainment processes and aerosol mixing affect the development of convective clouds.

We thank the reviewer for emphasizing the value of the comparison between the bin-microphysics parcel model and the observations, which underscores the importance of aerosol entrainment in the development of deep convective clouds. In the revised manuscript, we have expanded the introduction and discussion of entrainment effects. We agree that the extensive dataset obtained from individual flights offers valuable potential for developing general parameterizations that achieve improved agreement between simulations and observations. While this study focuses on case studies using a bin-microphysics parcel model to highlight the importance of incorporating entrainment, we are preparing a follow-up manuscript to conduct simulations for all flights during the DCMEX campaign. We will systematically investigate how different entrainment conditions (adiabatic, homogeneous, and inhomogeneous) affect not only cloud droplets but also secondary ice production. Regarding this, we have expanded the implications in the conclusion section.

**Regarding the above discussions, we have revised the manuscript in four aspects:**

1. We have re-organized the introduction to enhance the connection between the introduction and result sections. The revised introduction is:

[revised manuscript text omitted]

2. We have also expanded the discussion of entrainment processes in the development of convective clouds. In the revised manuscript, we examined three entrainment scenarios, including purely homogeneous mixing, purely inhomogeneous mixing, and a hybrid approach of early-stage inhomogeneous mixing (from the cloud base up to ~ 1 km above) followed by homogeneous mixing. We also examined the effects of aerosol entrainment under the three entrainment scenarios. By comparing different scenarios, a combination of early-stage inhomogeneous mixing and following homogeneous mixing as well as the inclusion of aerosol entrainment would provide the best match with observations. The revised manuscript in Sect. 3.2 is:

[revised manuscript text omitted]
 solid black lines and shades represent average and standard deviation of observed values. The dashed black, pink and green lines represent simulations under different scenarios respectively. The y-axis is the relative height with respect to the LCL height. It is noted that the pink and green dashed lines overlap in the plots of LWC.

Figure 8. Observed and simulated cloud droplet size distributions for the Period 1-C300 case (upper plots) and the Period 2-C303 case (bottom plots), as a function of T ranging from near cloud base to high level. The solid black lines and shades represent the average and standard deviation of observed values within specific T ranges. The dashed black, pink, and green lines represent simulations under different scenarios, respectively.

3. We have added the entrainment rate  $(\mu_j)$  to address the reviewer's comment of "no figure showing this entrainment parameter explicitly". We have added the exact handling and formulation of mixing with ambient air and secondary aerosol activation in the supplementary. The entrainment rate  $(\mu_j)$  is the key parameter for describing the entrainment process. Vertical profiles of simulated  $\mu_i$  are now shown in Fig. S8.

Figure S8. Figure S8. Vertical profiles of simulated  $\mu_j$  (m-1) for the Period 1-C300 case (left) and the Period 2-C303 case (right). The y-axis is the relative height with respect to the LCL height.

**4.** To expand the discussion of implications, the revised conclusion is:

"This study provides a valuable dataset for improving parameterizations of aerosol-cloud interactions and offers critical constraints for the accurate representation of microphysical processes in this area and similar deep convective systems for future model studies. The comparison between observations and bin-microphysics parcel model simulations also highlights the importance of incorporating entrainment processes from surrounding environments, particularly aerosol mixing, in future simulations of cloud droplets in deep convective systems. Future simulation studies should be conducted for all flight cases, to develop general parameterizations that achieve improved agreement between models and observations. In addition, Future studies should investigate how different entrainment conditions (adiabatic, homogeneous, and inhomogeneous) affect not only the development of cloud droplets but also ice production."

**Minor Comments**

Lines 40-43: no references are given. Please provide appropriate references.

In the re-organized introduction, this sentence has been deleted.

Lines 44-45: there are more references to add here, most notably the IPCC reports

References have been added.

Aerosols play a key role in these processes, and aerosol-cloud interactions are considered among the largest uncertainties in estimating climate sensitivity to radiative forcing (Boucher et al., 2013; Forster et al., 2021).

**Lines 54-55: again, no references given about the aerosol effect on cloud cover, lifetime, etc**

The introduction has been rephrased, and references have been added.

Generally, higher CCN concentrations produce a greater number of smaller droplets and narrower droplet size distributions, which are likely to inhibit collision-coalescence and delay raindrop formation, thereby extending cloud lifetime (Rosenfeld, 2000; Tao et al., 2012).

**Line 89: would it be nice to mention which advances in measurement techniques?**

The added manuscript is:

The campaign benefits from significant advances in measurement techniques, including high-resolution cloud probes with improved particle sizing and phase discrimination, and state-of-the-art aerosol instruments capable of resolving physicochemical properties (Finney et al., 2024).

**Line 111 and Line 114: please avoid using "etc"**

The "etc" has been deleted. The revised manuscript is:

The FAAM Bae-146 was equipped with a suite of instruments to measure atmospheric dynamics and thermodynamics, including such as wind speed and direction, temperature (T), and RH.

Ground-based meteorological stations provided such as surface T, RH, pressure, and precipitation intensity.

Line 178: cloud development → cloud parcel development? (since it is a cloud parcel model)

Accepted

Line 210: remains  $\rightarrow$  retains?

Accepted

Lines 292-294: no need to repeat the "(0.1 to 3 micron)" parenthesis, since  $N_a$  and  $N_{sa}$  were already defined earlier The "(0.1 to 3 micron)" has been deleted.

Line 310: please consider citing "Isoprene nitrates drive new particle formation in Amazon's upper troposphere" by Curtius et al. (2024). It is a recent and relevant paper on the subject, available in Nature: https://www.nature.com/articles/s41586-024-08192-4. Also note that the aerosol-rich layer in the Amazonian upper troposphere is mostly above 8 km, which seems to be the upper limit in your study.

The revised manuscript is:

Recent studies over the Amazon have highlighted the important role of extremely low-volatility organic compounds (e.g. Isoprene-derived organonitrates) formed from the oxidation of biogenic volatile organic compounds (VOCs) in driving this NPF process in upper troposphere (Zhao et al., 2020; Curtius et al., 2024). The observed ultrafine-particle-rich layer in Amazon's upper troposphere is mostly above 8 km, which is near the upper limit of the flight altitude range in this study.

Figure 7 (and others): there should be legends in the figures themselves to explain what the different lines are. The figures should be as self-sufficient as possible, without needed to read the caption.

The legends have been added into Figures 7 and 8.

**References**

[revised manuscript text omitted]